# Efficient Policy Optimization in Robust Constrained MDPs with Iteration Complexity Guarantees

**Sourav Ganguly**
Department of ECE
New Jersey Institute of Technology
New Jersey, USA
sg2786@njit.edu

**Kishan Panaganti**
Department of CMS
California Institute of Technology
(now at Tencent AI Lab, Seattle, WA)
kpb.research@gmail.com

**Arnob Ghosh**
Department of ECE
New Jersey Institute of Technology
New Jersey, USA
arnob.ghosh@njit.edu

**Adam Wierman**
Department of CMS
California Institute of Technology
California, USA
adamw@caltech.edu

## Abstract

Constrained decision-making is essential for designing safe policies in real-world control systems, yet simulated environments often fail to capture real-world adversities. We consider the problem of learning a policy that will maximize the cumulative reward while satisfying a constraint, even when there is a mismatch between the real model and an accessible simulator/nominal model. In particular, we consider the robust constrained Markov decision problem (RCMDP) where an agent needs to maximize the reward and satisfy the constraint against the worst possible stochastic model under the uncertainty set centered around an unknown nominal model. Primal-dual methods, effective for standard constrained MDP (CMDP), are not applicable here because of the lack of the strong duality property. Further, one cannot apply the standard robust value-iteration based approach on the composite value function either as the worst case models may be different for the reward value function and the constraint value function. We propose a novel technique that effectively minimizes the constraint value function–to satisfy the constraints; on the other hand, when all the constraints are satisfied, it can simply maximize the robust reward value function. We prove that such an algorithm finds a policy with at most $\epsilon$ sub-optimality and feasible policy after $O(\epsilon^{-2})$ iterations. In contrast to the state-of-the-art methods, we do not need to employ a binary search, thus, we reduce the computation time for larger value of discount factor ($\gamma$), and achieve a better performance for large state space.

## 1 Introduction

Ensuring safety or satisfying constraints is important for implementation of the RL algorithms in the real system. A poorly chosen action can lead to catastrophic consequences, making it crucial to incorporate safety constraints into the design. For instance, in self-driving cars [1], a slight safety violation can result in serious harm to the system. Constrained Markov Decision Process (CMDP) can address such safety concerns where the agent aims to maximize the expected reward while keeping the expected constraint cost within a predefined safety boundary [2] (cf.(5)). CMDPs effectively restricted agents from violating safety limits [3, 4]. However, in many practical problems, an algorithm is trained using a simulator which might be different from the real world. Thus, policies obtained for CMDP in simulated environment can still violate the constraint in the real environment.

To resolve the above issues, recently, researchers considered robust CMDP (RCMDP) problem where the constraint needs to be satisfied even when there is a model-mismatch due to the sim-

39th Conference on Neural Information Processing Systems (NeurIPS 2025).

| Wall-clock time comparisons (in secs) | | | | |
|---|---|---|---|---|
| **Environment Name** | **RNPG (our)** | **EPIRC-PGS** ($\gamma$ vals.) | | |
| | | 0.9 | 0.99 | 0.995 |
| **CRS** | 48.574 | 190.53 | 228.65 | 290.15 |
| **Garnet** | 78.406 | 290.21 | 316.23 | 453.14 |
| **Modified Frozenlake** | 160.31 | 453.7 | 561.47 | 620.12 |
| **Garbage collector** | 177.13 | 344.87 | 400.13 | 489.54 |

Table 1: Comparison of execution times averaged over multiple runs between RNPG, and EPIRC-PGS (inner loop T = 100 and outer-loop K=10) (Some more experimental results to demonstrate faster performance of our algorithms can be found in appendix G)

to-real gap. In particular, we seek to solve the problem

$$\textbf{RCMDP objective: } \min_{\pi}\max_{P\in\mathbb{P}} J_{c_0}^{\pi,P} \quad \text{s.t. } \max_{P\in\mathbb{P}} J_{c_i}^{\pi,P} \le b, \quad i \in \{1,\dots,K\}. \tag{1}$$

where $J_{c_n}$ is the expected cumulative cost for the associated RCMDP cost function $c_n$ (see Section 2). Here $\mathbb{P}$ is the uncertainty set centered around a nominal (simulator) model described in (6). Note that learning the optimal policy for RCMDP are more challenging compared to the CMDP. In particular, the main challenge lies in the fact that the standard primal-dual based approaches, which achieve provable sample complexity results for the CMDP problems [5, 6], cannot achieve the same for the robust CMDP problem as the problem may not admit strong duality even when the strict feasibility holds [7]. This is because the state occupancy measure is no longer convex as the worst-case transition probability model depends on the policy. Due to the same reason, even applying robust value iteration is not possible for the Lagrangian unlike the non-robust CMDP problem.

Recently, [8] proposed an epigraph approach to solve the problem in (1). In particular, they considered

$$\min_{\pi,b_0} \quad b_0 \quad \text{s.t. } J_{c_n}^{\pi} - b_n \le 0; \ n \in \{0,\dots,K\}. \tag{2}$$

Hence, the objective is passed on to the constraint with an objective of how tight the constraint can be. [8] finds the optimal policy for each $b_0$, and then optimized $b_0$ using a binary search. They showed that for each $b_0$, the iteration complexity is $O(\epsilon^{-4})$ to find the optimal policy. Note that one needs to evaluate robust value function at every iteration for each $b_0$ which is costly operation especially when $\gamma$ is large as it is evident by Table 1. Further, the binary search method only works when the estimation is perfect [9], thus, if the robust policy evaluator is noisy which is more likely for the large state-space, the binary search method may not work as it is evident in our function approximation setup (Appendix G). Moreover, the complexity of iteration is only $O(\log(\epsilon^{-1})\epsilon^{-4})$, which is worse than that of the CMDP [10]. We seek to answer the following:

> *Can we develop a computationally more efficient (without binary search) approach for robust CMDP problem with a faster iteration complexity bound?*

**Our Contributions**

- We propose a novel approach to address the optimization problem. Specifically, we reformulate it as follows:
$$\min_{\pi} \max \left\{ \frac{J_{c_0}^{\pi}}{\lambda}, \max_{n}\left[J_{c_n}^{\pi} - b_n\right] \right\}. \tag{3}$$

  This formulation balances the trade-off between optimizing the objective and satisfying the constraints. When $\max_n \left[J_{c_n}^{\pi} - b_n\right] > 0$, the focus is on reducing constraint violations. Otherwise, the objective $J_{c_0}^{\pi}$ is minimized, scaled by the factor $\lambda$. Notably, this framework eliminates the need for binary search over $\lambda$; solving the above problem directly yields a policy that respects the constraints for an appropriately chosen $\lambda$. We show the almost equivalence of optimal solution of (3) and (1). However, because of the point-wise maximum over the multiple objectives, it introduces additional challenges in achieving the iteration complexity, as the index of the value function of the objective now depends on the policy.
- We propose an algorithm (RNPG) that gives a policy which is at most $\epsilon$-sub optimal and feasible after $O(\xi^{-2}\epsilon^{-2})$ iterations if the strict feasibility parameter $\xi$ is known. This is the first result to show that strict safety feasibility can be achieved. This improves the existing iteration complexity

$O(\log\left(\frac{1}{(1-\gamma)\epsilon}\right)\epsilon^{-4})$ achieved by EPIRC-PGS by [8]. Our algorithm does not rely on binary search and uses KL regularization instead of projected gradient descent. We also show that if we do not know $\xi$, we can achieve a policy that violates the constraint by at most $\epsilon$ amount while being at most $\epsilon$-suboptimal with $O(1/\epsilon^4)$ iteration complexity. Moreover, our dependence on the state-space $(S)$, and the effective horizon (i.e., $\frac{1}{1-\gamma}$) are much better compared to EPIRC-PGS.

- We extend our framework to the function approximation setup by proposing a robust constrained actor-critic with integral probability metric as the uncertainty metric. For the finite-state, our empirical results show that our proposed approaches achieve a *feasible policy* with good reward (comparable or better than the one achieved by EPIRC-PGS, see Table 11) at a faster wall-clock time (see Table 1)[1] compared to the EPIRC-PGS. From Table 1, it is evident that our algorithm speeds up the computation process by at-least 2 times as compared to EPIRC-PGS algorithm when $\gamma = 0.9$ and at-least 3 times to EPIRC-PGS when $\gamma = 0.995$. For the function approximation setup, our proposed approach is the only one that achieves feasibility and even a better reward to the robust version of CRPO [11] during the test time for Cartpole experiment (Table 12). Further, we outperform EPIRC-PGS significant manner for the function approximation setup both in terms of performance and the training time showing its efficacy.

## 1.1 Related Works

**CMDP:** The convex nature of the state-action occupancy measure ensures the existence of a zero duality gap between the primal and dual problem for CMDP, making them well-suited for solution via primal-dual methods [2, 12–19]. The convergence bounds and rates of convergence for these methods have been extensively studied in [20–24, 6, 25, 26]. Beyond primal-dual methods, LP-based and model-based approaches have been explored to solve the primal problem directly [27, 18, 28, 29, 11, 30]. However, the above approaches cannot be extended to the RCMDP case.

**Robust MDP:** For robust (unconstrained) MDPs (introduced in [31]), recent studies obtain the sample complexity guarantee using robust dynamic programming approach [32–36]. Model-free approaches are also studied [34, 37–43]. However, extending these methods to Robust Constrained MDPs (RCMDPs) presents additional challenges. The introduction of constraint functions complicates the optimization process as one needs to consider the worst value function both for the objective and the constraint.

**RCMDP:** Unlike non-robust CMDPs, there is limited research available on robust environments. In [7], it was shown that the optimization function for RCMDPs is not convex, making it difficult to solve the Lagrangian formulation, unlike in standard CMDPs. Some studies have attempted to address this challenge using a primal-dual approach [44, 7] without any iteration complexity guarantee. [45] proposed a primal-dual approach to solve RCMDP under the strong duality by restricting to the categorical randomized policy class. However, they did not provide any iteration complexity guarantee. As we discussed, [8] reformulates the Lagrangian problem into an epigraph representation, addressing the limitations of previous methods while providing valuable theoretical insights. However, this method requires a binary search, significantly increasing computational complexity. Moreover, the binary search approach fails when the estimated robust policy value function is noisy [9].

## 2 Problem Formulation

**CMDP:** We denote a MDP as $\mathcal{M} = \langle \mathcal{S}, \mathcal{A}, \mathcal{P}, \mathcal{C}, \{c_j\}_{j=1}^K, \gamma \rangle$ where $\mathcal{S}, \mathcal{A}, \mathcal{P} : \mathcal{S} \times \mathcal{A} \times \mathcal{S} \to \mathbb{R}$ denote state space, action space, and probability transition function respectively. $\gamma \in [0, 1)$ denotes the discount factor and $c_i : \mathcal{S} \times \mathcal{A} \to \mathbb{R}$, for $i = \{0, 1, \ldots, K\}$, denotes the constraint function. Let $R_+ = \max(0, R)$ for any real number $R$ and $\pi : \mathcal{S} \to \mathcal{A}$ denote a policy. Let $\beta : \mathcal{S} \to \Delta(\mathcal{S})$ denote the initial state distribution where $\Delta(\mathcal{S})$ denotes the probability distribution taken over space $\mathcal{S}$. Let $V_{c_i}^{P,\pi}(s) : \mathcal{S} \to \mathbb{R}$, $s.t.$ $i \in \{0, \ldots, K\}$ (where $c_0 \in \mathcal{C}$ denote the cost for the objective) denote the value function obtained by following policy $\pi$ and the transition model $P$ where

$$V_{c_i}^{\pi,P}(s) := \mathbb{E}_{P,\pi}\left[\sum_{t=1}^{\infty} \gamma^{t-1}\pi(a|s)c_i^t(s,a)\right], \tag{4}$$

---

[1] The system specifications are, Processor: Intel(R)Core(TM)i7-14700-2.10 GHz, Installed RAM 32.0 GB (31.7 GB usable),64-bit operating system, x64-based processor No GPU.

where $c_i^t(s, a)$ denotes the single step '$i$'th-cost/reward for being at a state '$s$' and taking action '$a$' at the '$t$'-th instant. Without loss of generality, we assume $0 \leq c_i(s, a) \leq 1 \ s.t. \ i \in \{0, \ldots, K\}$. This is in consistent with the existing literature [8]. We also denote $J_{c_i}^{P,\pi} = \langle \rho, V_{c_i}^{P,\pi} \rangle$ for $i \in \{0, \ldots, K\}$ where $\rho$ is the initial state-distribution. *For notational simplicity, we denote $H = 1/(1 - \gamma)$ as the maximum cost value.*

The MDP $\mathcal{M}$ forms a constrained MDP when constraint cost functions are bounded by a threshold, leading to the following optimization problem,

$$\textbf{CMDP objective:} \quad \min J_{c_0}^{\pi,P} \ \text{s.t.} \ J_{c_i}^{\pi,P} \leq b_i \quad \forall i \in \{1, \ldots, K\}. \tag{5}$$

Note that even though we consider a cost-based environment to be consistent with the RCMDP literature [8] where the objective is to minimize the expected cumulative cost, our analysis can easily go through for reward-based environment where the objective is to maximize the expected cumulative reward. Further, we can also consider the constraints of the form $J_{c_i}^{\pi,P} \geq b_i$.

**RCMDP:** We consider that we have access to the nominal model $P_0$, however, the true model might be different compared to the nominal model $P_0$. Such a scenario is relevant when we train using simulator, however, the real environment might be different compared to the simulator. The state-of-the art choice for the uncertainty set is to collect all the probability distribution which are in close proximity to a nominal model $P_0 \in \Delta(\mathcal{S} \times \mathcal{A})$. Thus $\mathbb{P} = \bigotimes_{(s,a) \in \mathcal{S} \times \mathcal{A}} \mathbb{P}_{(s,a)}$ such that

$$\mathbb{P}_{(s,a)} = \{P \in \Delta(\mathcal{S}) : D(P, P_0(s, a)) \leq \rho\}, \tag{6}$$

where $D(., .)$ is the distance measure between two probability distribution and $\rho$ denotes the maximum perturbation possible from the nominal model. Some poplar choices for $D(., .)$ are TV distance, $\chi^2$ distance and KL-divergence [32].

Equation (6) satisfies the $(s, a)$-rectangularity assumption. It is important to note that our analysis and algorithm remain applicable as long as a robust policy evaluator, that is, $\max_{P \in \mathcal{P}} J_{c_i}^{\pi,P}$ is available. Therefore, we can also extend our approach to consider $s$-rectangular uncertainty sets. In addition, it is possible to extend this to the integral probability metric (IPM). However, without such an assumption, evaluating a robust value function becomes an NP-hard problem.

The objective in constrained robust MDPs is to minimize (or maximize in a reward based setting) the worst case value function while keeping the worst case expected cost function within a threshold (user defined) as defined in (1). *We denote $\max_P J_{c_i}^{P,\pi} = J_{c_i}^{\pi}$ as the worst possible expected cumulative cost corresponding to cost $c_i$ following the policy $\pi$.*

**Learning Metric**: Since we do not know the model, we are in the data-driven learning setting. Here, we are interested in finding the number of iterations ($T$) required to obtain a policy $\hat{\pi}$ with sub-optimality gap of at most $\epsilon$, and a feasible policy incurring no violations. That is, after $T$ iterations, $\hat{\pi}$ satisfies

$$\text{Gap}(\hat{\pi}) = J_{c_0}^{\hat{\pi}} - J_{c_0}^{\pi^*} \leq \epsilon \quad \text{and} \quad \text{Violation}(\hat{\pi}) = \max_n J_{c_n}^{\hat{\pi}} - b_n \leq 0, \tag{7}$$

where $\pi^*$ is the optimal policy of (1). Note that we do not assume any restriction on the policy class $\Pi$ unlike in [8]. In [45], the policy class increases as $T$ increases as it is an ensemble of the learned policies up to time $T$. Here, $\Pi$ denotes any Markovian policy.

Thus, the iteration complexity measures how many iterations required to obtain a feasible policy with sub-optimality gap of at most $\epsilon$. Iteration complexity is a standard measure for unconstrained robust MDP [38]. In addition to sub-optimality gap, we also seek to achieve a feasible policy $\hat{\pi}$ for RCMDP. Note that unlike in [8], where they allowed a violation of $\epsilon$, here, we want to find a feasible policy, a stricter requirement.

**Difficulty with the vanilla primal-dual method** The most celebrated method to solve a constrained optimization problem is by introducing Lagrangian multiplers. Let us consider $\lambda = (\lambda_1 \ldots \lambda_K) \in \mathbb{R}_+^N$ be the set of langrangian multipliers introduced to convert the primal problem eqn. (1) into the dual space which is shown in eqn. (8)

$$J^* = \min_{\pi \in \Pi} \max_{\lambda \in \mathbb{R}_+^N} \max_{P \in \mathbb{P}} J_{c_0}^{\pi,P} + \sum_{i=1}^{N} \lambda_i . \max_{P \in \mathbb{P}} (J_{c_i}^{\pi,P} - b_i). \tag{8}$$

In the CMDP problem, [46] shows that the strong duality holds when there exists a strictly feasible policy (aka Slater's condition). However, a concurrent work [47] highlighted that strong duality

does not hold for the RCMDP problem as the occupancy measure is no longer convex as the worst transition model differs for different policies. In addition to that, in [8] a strong ambiguity regarding the tractability in solving lagrangian problem is discussed. Further, even if one fixes $\lambda$, one cannot apply robust value iteration approach to find the optimal policy for the Lagrangian unlike the CMDP. Hence, it is evident to look for alternative measures to find a solution to the optimality problem.

# 3 Policy Gradient Approach for RCMDPs

In this section, we discuss our approach to solve the RCMDP problem (eqn.(1)). In what follows, we describe our policy optimization algorithm RNPG in detail.

## 3.1 Our Proposed Approach

In order to address the challenges of the primal-dual problem, We consider the following problem

$$\min_\pi \max\{J_{c_0}^\pi/\lambda, \max_n[J_{c_n}^\pi - b_n]\}. \tag{9}$$

**Intuition**: Note that when $J_{c_i}^\pi \leq b_i$ for all $i = 1, \ldots, K$, the second term in the objective becomes negative, and since $J_{c_0}^\pi \geq 0$, the optimization will focus on minimizing $J_{c_0}^\pi$, as the policy is likely to be feasible with respect to all constraints. Conversely, if there exists any $i$ such that $J_{c_i}^\pi > b_i$, then for a sufficiently large $\lambda$, the term $J_{c_0}^\pi/\lambda$ becomes smaller than $J_{c_i}^\pi - b_i$, causing the optimization to prioritize reducing the most violated constraint $J_{c_i}^\pi$.

Even though we can not claim that (9) and (1) are the same, we can claim that the optimal solution of (9) can only violate the constraint by at most $\epsilon$-amount by a suitable choice of $\lambda$. Hence, minimizing (9) amounts to searching for policies that can violate at most $\epsilon$ amount. Thus, the optimal policy of (1) can be an optimal of (9). In particular, optimal policy of (9) indeed has a smaller cost compared to that of (1). We formalize this as the following result.

**Proposition 1.** *Suppose that $\hat{\pi}^*$ is the optimal policy of (9) then $J_{c_0}^{\hat{\pi}^*} \leq J_{c_0}^{\pi^*}$, and can only violate the constraint by at most $\epsilon$ with $\lambda = 2H/\epsilon$.*

The key distinction from the epigraph-based approach proposed in [8] is that we avoid tuning the hyperparameter $b_0$ via binary search. This significantly reduces computational overhead, as also demonstrated in our empirical evaluations. Furthermore, tuning $b_0$ typically requires accurate estimation, even an unbiased estimation would not work, which is prohibitive as the state-space grows when a high-probability estimate becomes challenging.

Since our goal is to obtain a feasible policy, we assume that the optimal policy is strictly feasible.

**Assumption 1.** *We assume that $\max_n J_{c_n}^{\pi^*} - b_n \leq -\xi$, for some $\xi > 0$.*
The above assumption is required because we want to have a feasible policy rather bounding the violation gap to $\epsilon$. Note that we only need to know (or, estimate) the value of $\xi$. Of course, we do not need to know the optimal policy $\pi^*$. Using $\xi$, we can show that we achieve a *feasible* policy with at most $\epsilon$-gap in Theorem 4.1. Intuitively, if $\xi > \epsilon$, it means that by choosing $\lambda = 2H/\xi$, we can actually guarantee feasibility according to Proposition 1. We relax this $\xi$-dependency in Theorem 6.1 where we show that we can achieve a policy with at most $\epsilon$-gap and $\epsilon$-violation.

We consider the problem

$$\min_\pi \max\{J_{c_0}^\pi/\lambda, \max_n J_{c_n}^\pi - b_n + \xi\}. \tag{10}$$

Note that even though for theoretical analysis to achieve a feasible policy we assume the knowledge of $\xi$; for our empirical evaluations, we did not assume that and yet we achieved feasible policy with good reward exceeding the state-of-the-art performance. Hence, we modify the policy space to be $\xi$-dependent.

## 3.2 Policy Optimization Algorithm

We now describe our proposed robust natural policy gradient (RNPG) approach inspired from the unconstrained natural policy gradient [48]. For notational simplicity, we define $J_i(\pi) = J_{c_i}^\pi - b_i + \xi$ for $i = 1, \ldots, K$, and $J_0(\pi) = J_{c_0}^\pi/\lambda$. The policy update is then given by–

$$\pi_{t+1} \in \arg\min_{\pi \in \Pi} \langle \nabla_{\pi_t} J_i(\pi_t), \pi - \pi_t \rangle + \frac{1}{\alpha_t} \mathrm{KL}(\pi \| \pi_t) \quad \text{where } i = \arg\max\{\frac{J_{c_0}^\pi}{\lambda}, \{J_{c_n}^\pi - b_n + \xi\}_{n=1}^K\}$$

where KL is the usual Kullback-Leibler divergence. Note that this is a convex optimization problem, and can be optimized efficiently. If we use, $\ell_2$ regularization, i.e., $||\pi - \pi_t||_2^2$, then it becomes a robust projected policy gradient (RPPG) adapted from the unconstrained version (see Appendix D) [49, 38], a variant of which is used in [8] to find optimal policy for each $b_0$. Of course, our approach also works for $\ell_2$ norm which we define in Algorithm 4. Empirically, we observe that KL-divergence has a better performance, and provide iteration complexity for RNPG.

The complete procedure is described in Algorithm 1. First, we evaluate $J_{c_i}^{\pi_t}$ and $\nabla_{\pi_t} J_{c_i}^{\pi_t}$ using the robust policy evaluator which we describe in the following.

**Robust Policy Evaluator:** Our algorithm assumes access to a robust policy evaluation oracle that returns the worst-case performance of a given policy, i.e., $J_{c_i}^{\pi} = \max_{P \in \mathbb{P}} J_{c_i}^{\pi, P}$. This assumption is standard and is also adopted in both constrained [8] and unconstrained [38] robust MDP frameworks.

As we discussed, several efficient techniques exist for evaluating robust policies under various uncertainty models especially with $(s, a)$ rectangular assumption (6). In this work, we focus on the widely studied and expressive **KL-divergence-based uncertainty set**, which not only captures an infinite family of plausible transition models but also admits a closed-form robust evaluation method.

The robust value function under the KL-uncertainty set is formalized in Lemma D.1 (see Appendix D). The advantage is that we obtain a closed form expression for the robust value function, and we can evaluate it by drawing samples from the nominal model only. For further background on KL and other uncertainty sets, we refer the reader to [50, 34, 33].

Our framework is not limited to KL-divergence. Efficient robust value function evaluation techniques exist for other popular uncertainty models such as Total Variation (TV), Wasserstein, and $\chi^2$-divergence sets [32, 51, 52, 33]. These approaches typically leverage dual formulations to efficiently solve the inner maximization problem required for robust evaluation. We need our robust policy evaluator to be only $\epsilon$-accurate. For many uncertainty sets including popular $(s, a)$-rectangular perturbation (e.g., KL-divergence, TV-distance, $\chi^2$ uncertainty sets) this requires $O(1/\epsilon^2)$ samples [32, 34]. Hence, we need $T\epsilon^{-2}$ samples in those cases.

**Policy Update**: In order to evaluate $\nabla_{\pi_t} J_i^{\pi}$, we use the following result directly adapted to our setting from [48]

**Lemma 3.1.** *For any* $\pi \in \Pi$, *transition kernel* $P : S \times A \to \Delta(S)$, *for* $i = 1, \ldots, K$
$$(\nabla J_{i,P}(\pi))(s, a) = \frac{1}{1 - \gamma} d_P^{\pi}(s) Q_{i,P}^{\pi}(s, a), \text{ where } Q_{i,P}^{\pi}(s, a) = Q_{c_i, P}^{\pi}, \text{ and } Q_{0,P} = Q_{c_0, P}^{\pi}/\lambda.$$

Consider $i_t = \arg\max\{J_{c_0}^{\pi_t}/\lambda, \{J_{c_i}^{\pi_t} - b_i + \xi\}_{i=1}^K\}$, and $p_t = \arg\max J_{c_{i_t}}^{\pi_t, p_t}$, we can evaluate $\nabla_{\pi_t} J_{c_{i_t}}^{\pi_t, p_t}$ using the robust evaluator for $Q_{c_{i_t}}^{\pi_t, p_t}(\cdot, \cdot)$ as mentioned.

Hence, the natural policy update at iteration $t$ can be decomposed as multiple independent Mirror Descent updates across the states–

$$\pi_{t+1,s} = \arg\min_{\Delta_A}\{\langle Q_{i_t}^{\pi_t, p_t}, \pi_s \rangle + \frac{1}{\alpha_t} KL(\pi_s || \pi_{t,s})\}, \quad \forall s. \tag{11}$$

Again, this is efficient since it is convex. We use direct parameterization and soft-max parameterization for the policy update (Appendix F) by solving the optimization problem (11). The Algorithm 1 outputs $\pi_t^*$ corresponding to the minimum objective over $T$ iterations. We characterize $T$, the iteration complexity in the next section.

*Although Algorithm 1 includes $\xi$ for theoretical analysis, we do not assume knowledge of $\xi$ in our empirical evaluations. In Section 6, we discuss how we achieve a slightly weaker iteration complexity result without assuming the knowledge of $\xi$.*

# 4 Theoretical Results

In this section, we will discuss the results obtained for our RNPG algorithm (Algorithm 1). Before describing, the main results, we state the Assumptions.

**Assumption 2.** *There exists* $\beta \in (0, 1)$ *such that* $\gamma p(s'|s, a) \leq \beta p_0(s'|s, a) \, \forall s', s, a, \text{ and } p \in \mathcal{P}$.

This was a common assumption for unconstrained RMDP as well [53, 54]. Assumption 2 states that if the perturbed distribution assigns positive probability to an event, the nominal model should also

---

**Algorithm 1** Robust-Natural Policy Gradient for constrained MDP (RNPG)

---

**Input:** $\alpha$, $\lambda$, $T$, $\rho$, $\mathcal{V}(.)$ (Robust Policy Evaluator, see Algorithm 2))
**Initialize:** $\pi_0 = 1/|A|$.
**for** $t = 0 \ldots T - 1$ **do**
    $J_{c_i}^{\pi^t}, \nabla J_{c_i}^{\pi^t} = \mathcal{V}(c_i, \rho)$ where $i = \{0, 1, \ldots, K\}$.
    Update $\pi_{t+1}$ according to (11).
**end for**
Output policy $\arg\min_{\pi_t s.t. t \in \{0, \ldots, T-1\}} \max\{J_{c_0}^{\pi_t}/\lambda, \max_i(J_{c_i}^{\pi_t} - b_i + \xi)\}$.

---

assign positive probability to that event. Otherwise, a mismatch in supports could lead to unsampled regions and render finite-iteration bounds intractable. More importantly, Algorithm 1 does not need to know $\beta$. We also did not enforce in our empirical studies. The algorithm still performed well, suggesting that the practical impact may be less restrictive than the theory implies. Also, EPIRC-PGS [8] assumed that the ratio between the state-action occupancy measures on the states covered by all policies and the initial state distribution is bounded.

We also consider a slightly stronger optimal policy for the surrogate problem.

**Assumption 3.** *We consider $\hat{\pi}^*$, a uniform minimizer across all states of the surrogate problem in (9), i.e., $\hat{\pi}^*$ is a solution of $\min_\pi \max\{V_{c_0}^{\pi,P}(s)/\lambda, \max_n \max_P V_{c_i}^{\pi,P} - b_n\}$ for all $s$.*

A similar assumption is also considered for the unconstrained problem [54, 31].

**Theorem 4.1.** *Under Assumptions 1, 2, and 3 with $\lambda = 2H/\min\{\xi, 1\}$, $\alpha_t = \alpha_0 = \dfrac{1-\gamma}{\sqrt{TS}}$ after $T = O(\epsilon^{-2}\xi^{-2}(1-\gamma)^{-2}(1-\beta)^{-2}S\log(|A|))$ iterations, Algorithm 1 returns a policy $\hat{\pi}$ such that $J_r^{\hat{\pi}} - J_r^{\pi^*} \leq \epsilon$, and $\max_n J_{c_n}^{\hat{\pi}} - b_n \leq 0$.*

Thus, Algorithm 1 has an iteration complexity of $O(\epsilon^{-2})$. Also, the policy is feasible and only at most $\epsilon$-sub optimal. Our result improves upon the result $O(\epsilon^{-4})$ achieved in [8]. Further, they did not guarantee feasibility of the policy (rather, only $\epsilon$-violation). More importantly, we do not need to employ a binary search algorithm. Thus, our algorithm is computationally more efficient. Our dependence on $S$, $A$, and $(1-\gamma)^{-1}$ *are significantly better* compared to [8] as well. Note that for the unconstrained case the iteration complexity is $O(\epsilon^{-1})$ [48]; whether we can achieve such a result for robust CMDP has been left for the future.

As we mentioned, we do not use this $\xi$ for our empirical evaluations. Yet our results indicate that we can achieve feasible policy with better performance compared to EPIRC-PGS in significantly smaller time. In Section 6, we also obtain the result when we relax Assumption 1 with slightly worse iteration complexity by simply putting $\xi = 0$, and $\lambda = 2H/\epsilon$.

### 4.1 Proof Outline

The proof will be divided in two parts. First, we bound the iteration complexity for $\epsilon$-sub optimal solution of (10). Subsequently, we show that the sub-optimality gap and violation gap using the above result.

**Bounding the Iteration complexity for (10)**: The following result is the key to achieve the iteration complexity result of Algorithm 1 for the Problem (10)

**Lemma 4.2.** *The policy $\hat{\pi}$ returned by Algorithm 1 satisfies the following property:*

$$\max\{J_{c_0}^{\hat{\pi}}/\lambda, \max_n J_{c_n}^{\hat{\pi}} - b_n\} - \max\{J_{c_0}^{\pi}/\lambda, \max_n J_{c_n}^{\pi} - b_n\} \leq \epsilon/\lambda \tag{12}$$

*for any policy $\pi$ after $O(\lambda^2 \epsilon^{-2}(1-\gamma)^{-4}(1-\beta)^{-2})$ iterations under Assumptions 2, and 3.*

Hence, the above result entails that Algorithm 4 returns a policy which is at most $\epsilon$-suboptimal for the problem (10) after $O(\epsilon^{-2}\xi^{-2})$. We show that using this result we bound the sub-optimality and the violation gap.

**Technical Challenge**: The main challenge compared to the policy optimization-based approaches for unconstrained RMDP is that here the objective (cf.(9)) is point-wise maximum of multiple value

functions for a particular policy. Hence, one might be optimizing different objectives at different iterations at $\pi_t$ is varying across the iterations. Hence, unlike the unconstrained case, we cannot apply the robust robust performance difference Lemma as the value-function index might be different for $\pi_t$, and $\pi_{t+1}$. Instead, we bound it using Assumption 2, Holder's, and Pinsker's inequality.

**Bounding the Sub-optimality Gap**: By Assumption 1, $J_{c_n}^{\pi^*} \leq b_n - \xi$. Hence, $\max_n J_{c_n}^{\pi^*} - b_n + \xi \leq J_{c_0}^{\pi^*}/\lambda$ as $J_{c_0}^{\pi^*} \geq 0$. Thus,

$$(J_{c_0}^{\hat{\pi}} - J_{c_0}^{\pi^*})/\lambda$$
$$\leq \max\{J_{c_0}^{\hat{\pi}}/\lambda, \max_n J_{c_n}^{\hat{\pi}} - b_n + \xi\} - \max\{J_{c_0}^{\pi^*}/\lambda, \max_n J_{c_n}^{\pi^*} - b_n + \xi\} \leq \epsilon/\lambda \qquad (13)$$

where the last inequality follows from Lemma 4.2. By multiplying both the sides by $\lambda$, we have the result.

**Bounding the Violation**: We now bound the violations.

$$\max_n(J_{c_n}^{\hat{\pi}} - b_n + \xi) \leq \max_n(J_{c_n}^{\hat{\pi}} - b_n + \xi) - J_{c_0}^{\pi^*}/\lambda + H/\lambda$$
$$\leq \max\{J_{c_0}^{\hat{\pi}}/\lambda, \max_n J_{c_n}^{\hat{\pi}} - b_n\} - \max\{J_{c_0}^{\pi^*}/\lambda, \max_n J_{c_n}^{\pi^*} - b_n + \xi\} + H/\lambda$$
$$\leq \xi/2 + \epsilon/\lambda \leq \xi, \qquad (14)$$

where for the first inequality, we use the fact that $J_{c_0}^{\pi^*}/\lambda \leq H/\lambda$. For the secnond inequality, we use the fact that $J_{c_n}^{\pi^*} \leq b_n - \xi$. Hence, $\max_n J_{c_n}^{\pi^*} - b_n + \xi \leq J_{c_0}^{\pi^*}/\lambda$. Since $\lambda \geq 2H/\xi$, thus, $H/\lambda \leq \xi/2$. Note that $\epsilon/\lambda = \epsilon \min\{\xi, 1\}/(2H) \leq \epsilon\xi/(2H) \leq \xi/2$. Hence, the above shows that $\max_n(J_{c_n}^{\hat{\pi}} - b_n) \leq 0$.

## 5 Experimental Results

We evaluate our algorithms[2] on two environments: (i) *Garnet*, and (ii) *Constrained Riverswim (CRS)*. Additional experimental results are provided in Appendix E.

We have fixed $\lambda = 50$ across the environments. This demonstrates that with the inclusion of a KL regularization term over policy updates, RNPG eliminates the need for manual tuning of $\lambda$. A sufficiently large fixed value ($\lambda = 50$) yields consistently strong performance. For a detailed description of the hyperparameters used, please refer to Appendix E.

**Garnet:** The Garnet environment is a well-known RL benchmark that consists of $nS$ states and $nA$ actions, as described in [55]. For our experiments, we consider $\mathcal{G}(15, 20)$ with 15 states and 20 actions. The *nominal probability function*, *reward function*, and *utility function* are each sampled from separate normal distributions: $\mathcal{N}(\mu_a, \sigma_a)$, $\mathcal{N}(\mu_b, \sigma_b)$, and $\mathcal{N}(\mu_c, \sigma_c)$, where the means $\mu_a, \mu_b$, and $\mu_c$ are drawn from a uniform distribution $Unif(0, 100)$. To ensure valid probability distributions, the nominal probabilities are exponentiated and then normalized. In this environment, we seek to maximize the reward while ensuring that the constraint is above a threshold.

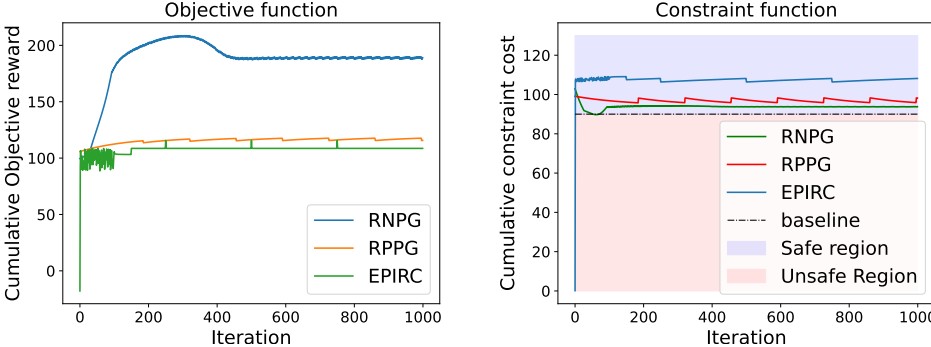

Figure 1: Comparison of RNPG, RPPG and EPIRC-PGS on Garnet(15,20) environment. Here, we want to maximize the objective (vf), and want the constraint (cf) to be above the baseline.

**Constrained River-swim (CRS):** The River-Swim environment consists of six states, each representing an island in a water body. The swimmer begins at any island and aims to reach either end

[2]The complete implementation is available at `https://github.com/Sourav1429/RCAC_NPG.git`

of the river to earn a reward. At each state, the swimmer has two possible actions: *swim left* ($a_0$) or *swim right* ($a_1$). Rewards are only provided at the boundary states, while intermediate states do not offer any rewards. The leftmost state, $s_0$, and the rightmost state, $s_5$, correspond to the riverbanks. As the swimmer moves from $s_0$ to $s_5$, the water depth increases, and dangerous whirlpools become more prevalent. This progression is captured by a *safety constraint cost*, which varies across states. The safety cost is lowest at $s_0$ and reaches its maximum at $s_5$, reflecting the increasing risk as the swimmer ventures further downstream. Here the goal is to maximize the cumulative reward while ensuring the cumulative cost is below a threshold.

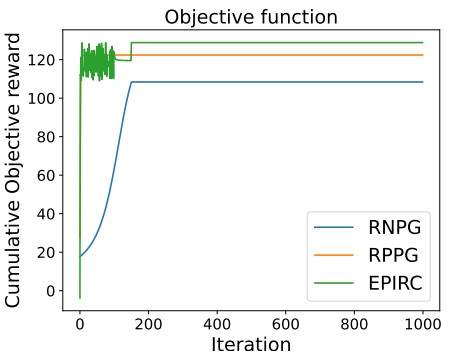 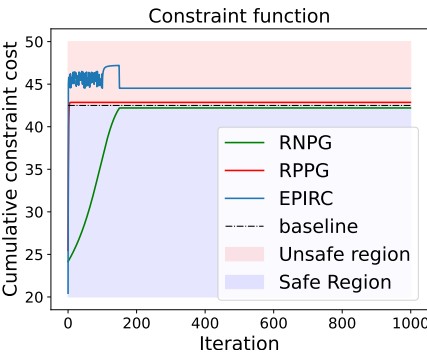

Figure 2: Comparison of RPPG and EPIRC-PGS on CRS environment. Here we want to maximize the objective (vf) while constraint (cf) being below the threshold line.

## 5.1 Analysis of results

- *Does RNPG perform better than EPIRC-PGS?*

  **Performance**: Our experimental results demonstrate that RNPG consistently outperforms EPIRC-PGS, in both environments. *In fact, for the CRS environment (Figure 2) RNPG is the only one that produces a feasible policy*. EPIRC-PGS is unable to produce a feasible policy there. In the Garnet environment (Figure 1), RNPG finds a feasible policy while achieving a better reward compared to the EPIRC-PGS. Also, RNPG shows a better convergence property and is more stable because of the KL regularization.

  **Computational Time**: RNPG exhibits significant improvements in computational efficiency, achieving convergence at least 4x faster than EPIRC-PGS for $\gamma = 0.9$, and 6x faster for $\gamma = 0.995$ in the CRS setting (Table 1). In the Garnet environment, RNPG achieves a 3x speedup over EPIRC-PGS for $\gamma = 0.9$, and at least 5x speedup for $\gamma = 0.995$ (Table 1). The difference in runtime can be attributed to the fact that RNPG eliminates the need for binary search for each $b_0$ value in (2) as described above, and it uses a KL regularization.

  To summarize, RNPG performs better compared to EPIRC-PGS in terms of achieving a better reward while maintaining feasibility across the environments. Moreover, the convergence is stable across the environments, and reduces the computational time significantly compared to EPIRC-PGS as theoretical result suggested.

- *KL regularization compared to $\ell_2$ regularization.*

  We also compare RNPG with RPPG (see Appendix D), a projected robust gradient descent variant that uses an $\ell_2$ regularizer instead of KL for policy update in (11). In the CRS environment, RPPG performs slightly better than EPIRC-PGS by maintaining smaller constraint violations, though it still occasionally breaches the safety threshold. In the Garnet environment, RPPG achieves a better performance compared to EPIRC-PGS while maintaining feasibility, however, it achieves a smaller reward compared to RNPG. RNPG is also much stable, showing that KL regularization is more effective compared to $\ell_2$ regularization. We observe that RPPG (and, similar to RNPG) also has a smaller computational time compared to EPIRC-PGS, which demonstrates that removing the binary search is the key, as EPIRC-PGS also uses $\ell_2$ regularization for policy update.

- *Does $\lambda$ require extensive tuning for RNPG?*

  A particularly notable observation from our experiments is that RNPG performs robustly across different environments using a fixed value of $\lambda = 50$. This highlights that RNPG does not need to set different $\lambda$ values for different environments as theoretical result suggested. Rather, one high $\lambda$-value is enough to achieves feasibility while achieving good reward.

# 6 Discussions and Limitation

**Relaxing Assumption 1**: We achieve our results in Theorem 4.1 where we assume that the optimal policy is strictly feasible and the feasibility parameter $\xi$ is known. We will relax both the features of the assumption that $\xi$ is known, and the optimal policy is strictly feasible in the following with a slightly worse iteration complexity while ensuring that the policy has violation of at most $\epsilon$, the same metric achieved by EPIRC-PGS [8].

**Theorem 6.1.** *Algorithm 1 gives a policy $\hat{\pi}$ such that $J_{c_0}^{\hat{\pi}} - J_{c_0}^{\pi^*} \leq \epsilon$ and $\max_n J_{c_n}^{\hat{\pi}} - b_n \leq \epsilon$ after $O(\epsilon^{-4}(1-\gamma)^{-4}(1-\beta)^{-2}\log(|A|))$ number of iterations when we plug $\lambda = 2H/\epsilon$ and $\xi = 0$.*

Note that since we are not assuming strict feasibility of the optimal policy, we can only bound the violation up to $\epsilon$. The key here is to use $\lambda = 2H/\epsilon$ as we do not know $\xi$, and then obtain an $\epsilon^2$-close result using Lemma 4.2. This makes the iteration complexity of $O(\epsilon^{-4})$. Note that our dependence on $S$, $A$, and $1/(1-\gamma)$ are *significantly better* compared to EPIRC-PGS[8]. Further, we do not employ binary search. The proof is in Appendix C.

## 6.1 Extending to Function Approximation: Robust Constrained Actor-Critic (RCAC)

We extend our framework to the function approximation setting motivated by the work of [54] for unconstrained Robust MDP problem. In particular, we consider the integral probability metric (IPM) as an uncertainty set, $d_{\mathcal{F}}(p,q) = \sup_{f \in \mathcal{F}}\{p^T f - q^T f\}$ where $\mathcal{F} \subseteq \mathbb{R}^{|S|}$. Many metrics such as Kantorovich metric, total variation, etc., are special cases of IPM under different function classes [56]. We then consider the IPM-based uncertainty set, $\mathcal{P}_{s,a} = \{q | d_{\mathcal{F}}(q, p_{s,a}^0) \leq \rho\}$ around the nominal model.

Consider the linear function approximation setting where $V_{c_i,w}^{\pi} = \Psi w_{c_i}$ where $\Psi \in \Re^{|S| \times d}$ is a feature matrix of $\psi^T(s) \ \forall s$ as each row. We now consider the following function class $\mathcal{F} = \{s \to \psi(s)^T \zeta, \zeta \in \Re^d, ||\zeta||_2 \leq 1\}$. Now, we can apply the Proposition 1 in [57] to achieve the worst case value function. In particular, we have $\sup_{q \in d_{\mathcal{F}}(q, p_{s,a}^0)} q^T V_{c_i,w}^{\pi} = (p_{s,a}^0)^T V_{c_i,w}^{\pi} + \rho||w_{c_i,2:d}^{\pi}||_2$ where we normalize to let the first coordinate of $\psi(s) = 1$. Hence, we can use the following equation to compute the robust Bellman operator

$$L_{\mathcal{P}} V_{c_i,w}^{\pi} = c_i(s,a) + \gamma V_{c_i,w}^{\pi} + \rho||w_{c_i,2:d}||_2, \tag{15}$$

with the next state $s'$ is drawn from the nominal model. Guided by the last regularization term of the empirical robust Bellman operator in (15), when considering value function approximation by neural networks we add a similar regularization term for all the neural network parameters except for the bias parameter in the last layer. We use the expression in (15) for the robust value function for the gradient, and $J_{c_i}$ in Algorithm 1. We need to estimate the robust Q function. In order to estimate the $Q$-function we first use (15) by plugging the $V$-approximation, and then we use the linear regression to fit the critic for the robust $Q$-function. The details can be found in Appendix G.

From the empirical results in Figure 9 and Table 12, it is evident that our proposed approach outperforms the state-of-the-art approaches. More importantly, compared to the EPIRC-PGS (adapted to the function approximation setting), our approach achieves a significantly better performance with small wall-clock time. Also, our proposed approach outperforms the robust version of the CRPO algorithm [11, 47] and achieves feasibility unlike robust CRPO. In Appendix H we showed that robust CRPO may not achieve a finite time iteration complexity guarantee even for finite-state space.

# 7 Conclusions and Future Works

In this work, we present a novel algorithm that leverages the projected policy gradient and natural policy gradient techniques to find an $\epsilon$-suboptimal and a feasible policy after $O(\epsilon^{-2})$ iterations for RCMDP problem. We demonstrate the practical applicability of our algorithm by testing it on several standard reinforcement learning benchmarks. The empirical results highlight the effectiveness of RNPG, particularly in terms of reduced computation time and achieving feasibility and a better reward compared to other state-of-the-art algorithms for RCMDP.

Relaxing Assumption 2, and 3 constitute an important future research direction. Achieving a lower bound or improving the iteration complexity is also an important future research direction. Characterizing the results for other uncertainty sets also constitutes an important future research direction. Iteration complexity guarantee for the function approximation setting has been left for the future.

## Acknowledgments

AG and SG acknowledge NJIT Startup Fund indexed 172884. SG acknowledges Neurips 2025 for awarding him with the NeurIPS 2025 Scholar Award. AW and KP acknowledge NSF grants CCF-2326609, CNS-2146814, CPS-2136197, CNS-2106403, and NGSDI-2105648 and support from the Resnick Sustainability Institute. KP also acknowledges support from the 'PIMCO Postdoctoral Fellow in Data Science' fellowship at the California Institute of Technology and the Resnick Institute.

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

# Contents

# A  Proof of Proposition 1

*Proof.* Recall that $\hat{\pi}^*$ is the solution of (9). For the first result, we have:

$$J_{c_0}^{\hat{\pi}^*}/\lambda - J_{c_0}^{\pi^*}/\lambda \le \max\{J_{c_0}^{\hat{\pi}^*}/\lambda, \max_n[J_{c_n}^{\hat{\pi}^*} - b_n]\} - \max\{J_{c_0}^{\pi^*}/\lambda, \max_n[J_{c_n}^{\pi^*} - b_n]\}$$
$$\le 0 \tag{16}$$

where we use the fact that $\pi^*$ is feasible in the first inequality. For the second inequality, we use the optimality of $\hat{\pi}^*$ for (9).

We prove the second result using contradiction. Assume that the optimal solution $\hat{\pi}^*$ of (9) violates the constraint by $\epsilon$. We then show by contradiction that it cannot be an optimal solution of (9). Since at least one of the constraints violates by $\epsilon$, thus $\max_n[J_{c_n}^{\hat{\pi}^*} - b_n] \ge \epsilon$. Note that since $\lambda = 2H/\epsilon$, therefore $J_{c_0}^{\hat{\pi}^*} \le \epsilon/2$ as the maximum value of $J_{c_0}^{\hat{\pi}^*}$ is $H$. Thus, we have

$$\max\{J_{c_0}^{\hat{\pi}^*}/\lambda, \max_n[J_{c_n}^{\hat{\pi}^*} - b_n]\} \ge \epsilon. \tag{17}$$

Now, consider the optimal solution $\pi^*$ of (1). It is feasible thus $\max_n[J_{c_n}^{\pi^*} - b_n] \le 0$. Further, $J_{c_0}^{\pi^*}/\lambda \le \epsilon/2$. Hence,

$$\max\{J_{c_0}^{\pi^*}/\lambda, \max_n[J_{c_n}^{\pi^*} - b_n]\} \le \epsilon/2 < \epsilon \le \max\{J_{c_0}^{\hat{\pi}^*}/\lambda, \max_n[J_{c_n}^{\hat{\pi}^*} - b_n]\}, \tag{18}$$

which contradicts the fact that $\hat{\pi}$ is optimal for (9). This proves the second result. $\qquad\square$

# B  Proof of Lemma 4.2

We use the following results proved in [48] in order to prove Lemma 4.2.

**Lemma B.1** ([48, Lemma 4.1]). *Let us assume that* $i_t = \arg\max\{J_{c_i}^{\pi_t} - b_i\}$, *then,*

$$\alpha_t \langle Q_{s,i_t}^{\pi_t, p_t}, \pi_{t+1,s} - y\rangle + B(\pi_{t+1,s}, \pi_{t,s}) \le B(y, \pi_{t,s}) - B(y, \pi_{t+1,s}), \tag{19}$$

*where $B$ is the Bregman's divergence.*

The above result follows from Bregmen divergence and the policy update. In our case, $B$ is the KL divergence.

**Lemma B.2** ([48, Lemma A.3]). *For any $\pi, \pi'$, $p$, $c_i$, and $\rho$, we have*

$$J_{c_i,\rho}(\pi', p) - J_{c_i,\rho}(\pi, p) = \frac{1}{1-\gamma}\sum_s d_\rho^{\pi,p}(s)\sum_a (\pi'_{s,a} - \pi_{s,a})Q_{s,a,c_i}^{\pi',p}. \tag{20}$$

The following result is a direct consequence of Assumption 2, and has been proved in Appendix B.1.

**Lemma B.3.** *For any $\pi \in \Pi$*

$$\Phi(\pi) - \Phi(\hat{\pi}^*) \le \frac{1}{1-\beta}\mathbb{E}_{s\sim d_\rho^{\pi^*, p_0}}[\langle Q_{c_i}^{\pi, p}, \pi_s - \hat{\pi}^*\rangle], \tag{21}$$

*where $i = \arg\max\{J_{c_i}^{\pi} - b_i\}$, and $p = \arg\max J_{c_i}^{\pi, P}$.*

Now, we are ready to prove Lemma 4.2.

*Proof.* From Lemma B.3

$$\Phi(\pi_t) - \Phi(\hat{\pi}^*) \le \frac{1}{1-\beta}\sum_s d_\rho^{\hat{\pi}^*, p_0}(s)\sum_a (\pi_{t,s,a} - \hat{\pi}_{s,a}^*)\hat{Q}_{s,a,c_{i_t}}^{\pi_t, p_t}.$$

where $\hat{Q}$ is the estimated value. Consider that the worst-case evaluator is only $\epsilon_0$ is close that is $||Q_{c_{i_t}}^{\pi_t, p_t} - \hat{Q}_{c_{i_t}}^{\pi_t, p_t}||_\infty \le \epsilon_0$.

Hence, we have

$$\Phi(\pi_t) - \Phi(\hat{\pi}^*) \le \frac{1}{1-\beta} \sum_s d_\rho^{\hat{\pi}^*, p_0}(s) \sum_a (\pi_{t,s,a} - \hat{\pi}_{s,a}^*) Q_{s,a,c_{i_t}}^{\pi_t, p_t} + \epsilon_0 \tag{22}$$

Applying Lemma B.1 (subtracting and adding $\langle Q_{s,i_t}^{\pi_t,p_t}, \pi_{t,s} \rangle$ , we then have from (22)

$$\Phi(\pi_t) - \Phi(\hat{\pi}^*) \le$$
$$[\frac{1}{1-\beta}\mathbb{E}_{s\sim d_\rho^{\hat{\pi}^*,p_0}}[\langle Q_{s,i_t}^{\pi_t,p_t}, \pi_{t,s} - \pi_{t+1,s} \rangle + \frac{1}{\alpha}B(\hat{\pi}^*, \pi_t) - \frac{1}{\alpha}B(\hat{\pi}^*, \pi_{t+1}) - \frac{1}{\alpha}B(\pi_{t+1}, \pi_t)]] + \epsilon_0. \tag{23}$$

Now,

$$\langle Q_{s,i_t}^{\pi_t,p_t}, \pi_{t,s} - \pi_{t+1,s} \rangle - \frac{1}{\alpha}B(\pi_{t+1}, \pi_t) \le ||q_{s,i_t}^{\pi_t,p_t}||_\infty ||\pi_{t,s} - \pi_{t+1,s}||_1 - \frac{1}{2\alpha}||\pi_{t,s} - \pi_{t+1,s}||_1^2$$
$$= \frac{-1}{2\alpha}(\alpha_t||Q_{s,i_t}^{\pi_t,p_t}||_\infty - ||\pi_{t,s} - \pi_{t+1,s}||_1)^2 + \frac{\alpha}{2}||Q_{s,i_t}^{\pi_t,p_t}||_\infty^2$$
$$\le \frac{\alpha}{2}||Q_{s,i_t}^{\pi_t,p_t}||_\infty^2. \tag{24}$$

where we use the Holder's inequality for the first inequality. For the second inequality, we use the Pinsker's inequality as $B$ is the KL divergence.

Hence, by summing over $t$, and using (24) we have from (23),

$$\sum_t (\Phi(\pi_t) - \Phi(\hat{\pi}^*)) \le \frac{1}{1-\beta}\sum_{t=0}^{T-1}\mathbb{E}_{s\sim d_\rho^{\hat{\pi}^*,p_0}}\alpha||Q_{s,i}^{\pi_t,p_t}||_\infty^2$$
$$+ \frac{1}{\alpha(1-\beta)}\mathbb{E}_{s\sim d_\rho^{\hat{\pi}^*,p_0}}[B(\hat{\pi}^*, \pi_t) - B(\hat{\pi}^*, \pi_{t+1})] + T\epsilon_0$$
$$\le \frac{1}{(1-\beta)}\sum_{t=0}^{T-1}\alpha S \frac{1}{(1-\gamma)^2} + \frac{1}{\alpha(1-\beta)}\mathbb{E}_{s\sim d_\rho^{\hat{\pi}^*,p_0}}B(\hat{\pi}^*, \pi_0) + T\epsilon_0. \tag{25}$$

Here, we use the fact that $||Q_{s,a,c_i}^{\pi_t,p_t}||_\infty \le \frac{1}{1-\gamma}$. This is easy to discern for $i = \{1,\ldots,K\}$. For $i = 0$, we have $||Q_{s,a,c_0}^{\pi_t,p_t}||_\infty \le \frac{1}{(1-\gamma)\lambda} \le \min\{\xi/2, 1/2\} < \frac{1}{1-\gamma}$ as $\xi \le \frac{1}{1-\gamma}$. Hence, from (25),

$$\sum_t (\Phi(\pi_t) - \Phi(\hat{\pi}^*)) \le \frac{1}{(1-\beta)}T\alpha S\frac{1}{(1-\gamma)^2} + \frac{1}{\alpha(1-\beta)}\mathbb{E}_{s\sim d_\rho^{\hat{\pi}^*,p_0}}B(\hat{\pi}^*, \pi_0) + T\epsilon_0. \tag{26}$$

Now, replacing $\pi_0 = \frac{1}{|A|}$, and $\alpha = \frac{(1-\gamma)}{\sqrt{TS}}$, we have

$$\sum_t (\Phi(\pi_t) - \Phi(\hat{\pi}^*)) \le \frac{1}{1-\beta}\sqrt{ST}\log(|A|)\frac{1}{\sqrt{(1-\gamma)^2}} + T\epsilon_0. \tag{27}$$

Thus,

$$\Phi(\hat{\pi}) - \Phi(\hat{\pi}^*) \le \frac{1}{T}\sum_t (\Phi(\pi_t) - \Phi(\hat{\pi}^*)) \le \frac{\sqrt{S}\log(|A|)}{(1-\beta)\sqrt{T(1-\gamma)^2}} + \epsilon_0, \tag{28}$$

where we use the fact that $\hat{\pi} = \arg\min_{t=0,\ldots,T-1}\max\{J_{c_0}^{\pi_t}, \max_n\{J_{c_n}^{\pi_t} - b_n\}\}$.

Hence, when $T = O(\frac{4}{(1-\gamma)^2(1-\beta)^2}S\log(|A|)(1/\epsilon^2))$ iteration, the above is bounded by $\epsilon$, if $\epsilon_0 = \epsilon/2$. The result now follows. □

## B.1 Proof of Lemma B.3

*Proof.* Let $i = \arg\max\{J_{c_i}^\pi - b_i\}$, and $p = \arg\max J_{c_i}^{\pi,P}$. Now,

$$\Phi(\pi) - \Phi(\hat{\pi}^*) \leq J_{c_i}^{\pi,p} - \max_P J_{c_i}^{\hat{\pi}^*,P}. \tag{29}$$

We now bound the right-hand side, and assume that $p^* = \arg\max J_{c_i}^{\hat{\pi}^*,P}$

$$V_{c_i}^\pi(\rho) - V_{c_i}^{\hat{\pi}^*}(\rho) = V_{c_i}^\pi(\rho) - \mathbb{E}_{s\sim\rho}\mathbb{E}_{\hat{\pi}^*}[c_i(s,a) + \gamma \sum_{s'} p^*(s'|s,a)V_{c_i}^{\pi^*}(s')]$$

$$= \mathbb{E}_{s\sim\rho}\mathbb{E}_{a\sim\hat{\pi}^*}[V_{c_i}^\pi(\rho) - c_i(s,a) + \gamma \sum_{s'} p(s'|s,a)V_{c_i}^\pi(s')]$$

$$- \mathbb{E}_{s\sim\rho}\mathbb{E}_{\hat{\pi}^*}\gamma[\sum_{s'} p^*(s'|s,a)V_{c_i}^{\hat{\pi}^*}(s') - \sum_{s'} p(s'|s,a)V_{c_i}^\pi(s')]$$

$$\leq \sum_{s\sim\rho}\langle Q_{c_i}^\pi, \pi - \hat{\pi}^*\rangle - \gamma\mathbb{E}_{s\sim\rho}\mathbb{E}_{\hat{\pi}^*}[\sum_{s'} p(s'|s,a)(V_{c_i}^{\hat{\pi}^*}(s') - V_{c_i}^\pi(s'))]$$

where the inequality follows from the fact that $p^* = \arg\max \sum_{s'} p^*(s'|s,a)V_{c_i}^{\hat{\pi}^*}(s')$. Hence,

$$V_{c_i}^\pi(\rho) - V_{c_i}^{\hat{\pi}^*}(\rho) \leq \sum_{s\sim\rho}\langle Q_{c_i}^\pi, \pi - \hat{\pi}^*\rangle + \beta\mathbb{E}_{s\sim\rho}\mathbb{E}_{\hat{\pi}^*}[\sum_{s'} p_0(s'|s,a)(V_{c_i}^\pi(s') - V_{c_i}^{\hat{\pi}^*}(s'))]. \tag{30}$$

where we use the fact that $V_{c_i}^\pi(s') \geq V_{c_i}^{\hat{\pi}^*}(s')$ by Assumption 3, and Assumption 2. By recursively, expanding we get the result. $\square$

## C  Proof of Theorem 6.1

*Proof.* Here, we just consider the following objective

$$\min_\pi \max\{J_{c_0}^\pi/\lambda, \max_n J_{c_n}^\pi - b_n\}, \tag{31}$$

since we do not know $\xi$, here, we only use $\max_n J_{c_n}^\pi - b_n$ instead of $\max_n J_{c_n}^\pi - b_n + \xi$. We consider $\lambda = 2H/\epsilon$.

**Sub-optimality gap**: Since $\pi^*$ is feasible, thus, $J_{c_0}^{\pi^*}/\lambda \geq \max_n J_{c_n}^{\pi^*} - b_n$. Thus,

$$(J_{c_0}^{\hat{\pi}} - J_{c_0}^{\pi^*})/\lambda$$
$$\leq \max\{J_{c_0}^{\hat{\pi}}/\lambda, \max_n J_{c_n}^{\hat{\pi}} - b_n\} - \max\{J_{c_0}^{\pi^*}/\lambda, \max_n J_{c_n}^{\pi^*} - b_n\}$$
$$\leq \epsilon^2/(2H), \tag{32}$$

where the inequality follows from Lemma 4.2 with $\lambda = O(1/\epsilon)$. Now, using $\lambda = 2H/\epsilon$ and multiplying both the sides we get the results.

**Violation Gap**: We now bound the violation.

$$\max_n J_{c_n}^{\hat{\pi}} - b_n$$
$$\leq \max\{J_{c_0}^{\hat{\pi}}/\lambda, \max_n J_{c_n}^{\hat{\pi}} - b_n\} - \max\{J_{c_0}^{\pi^*}/\lambda, \max_n J_{c_n}^{\pi^*} - b_n\} + H/\lambda$$
$$\leq \epsilon^2/(2H) + \epsilon/2 \leq \epsilon, \tag{33}$$

where we use the fact that $J_{c_0}^{\pi^*}/\lambda \leq H/\lambda \leq \epsilon/2$. Hence, the result follows. $\square$

## D  Robust policy evaluator based on KL divergence

**Robust Policy evaluator:** Our algorithm assumes the existence of a robust policy evaluator oracle that evaluates $\max_{P\in\mathbb{P}} J_{c_i}^{\pi,P}$ for a given $\pi$. There are many evaluation techniques that are used to

efficiently evaluate a robust policy perturbed by popular uncertainty measures. In this work, we evaluate our policies using a variant EPIRC-PGS algorithm [8] for KL uncertainty set (as shown in Algorithm 2).

The general robust DP equation is given by Equation (34)

$$\textbf{(ROBUST DP):} \quad Q_{c_n}^{(t+1)}(s,a) = c_n(s,a) + \gamma \max_{p \in \mathbb{P}} \sum_{s' \in \mathcal{S}} p(s') V_{c_n}^t(s'),$$

$$\text{where } V_{c_n}^t(s') := \sum_{a' \in \mathcal{A}} \pi(s',a') Q_{c_n}^t(s',a'). \tag{34}$$

$$\mathbb{P} = \otimes_{s,a} \mathbb{P}_{(s,a)} \quad \text{where} \quad \mathbb{P}_{(s,a)} = \{P \in \Delta(\mathcal{S}) | \text{KL}\,[P|P_0(.|s,a)] \leq C_{KL}\},$$

where $\mathbb{P}$ satisfies $(s,a)$-rectangularity assumption and $\text{KL}[p|q] = \sum_{s \in \mathcal{S}} p(s) \ln \frac{p(s)}{q(s)}$ for two probability distribution $p, q \in \Delta(\mathcal{S})$. The KL uncertainty evaluator (see Algorithm 2) is justified by Lemma 4 and 5 in [8].

---

**Algorithm 2** KL Uncertainty Evaluator

---

1: **Input:** policy $\pi$, nominal probability transition function $p^0$, perturbation parameter $C_{KL}$, $c_i = [c_0, c_1, \ldots c_K]$, discount factor $\gamma$, $\rho$, $|\mathcal{S}|$, $|\mathcal{A}|$
2: $Q, V = \text{Robust\_Q-table}(c_i, \pi, p^0, C_{KL})$ (see Algorithm 3)
3: $P^*[s,a,.] = \dfrac{p^0[s,a,.]\exp\left(\frac{V[.]}{C_{KL}}\right)}{\sum_{s' \in \mathcal{S}} p^0[s,a,s']\exp\left(\frac{V[s']}{C_{KL}}\right)} \quad \forall (s,a) \in \mathcal{S} \times \mathcal{A}$
4: $T[s,s'] = \sum_{a \in \mathcal{A}} \pi(a|s) P^*(s,a,s'), \quad \forall (s,s') \in \mathcal{S} \times \mathcal{S}$
5: $Q_{c_i,P^*}^\pi = (I - \gamma T)^{-1} c_i$
6: $\hat{J} = \rho^T \left( \sum_{a \in \mathcal{A}} (\pi(a|s) Q_{c_i,P^*}^\pi(s,a)) \right)$
7: $d_{P^*}^\pi = (1 - \gamma)(I - \gamma T)^{-1}\rho$
8: $\nabla \hat{J} = H d_{P^*}^\pi(s) Q_{c_i,P^*}^\pi(s,a) \ \forall \ (s,a) \in \mathcal{S} \times \mathcal{A}$
9: **Return:** $\hat{J}, \nabla \hat{J}$

---

The KL uncertainty evaluator follows from Lemma D.1. In Algorithm 2, we need the Robust_Q-table. The compact algorithm for that is given in Algorithm 3.

**Lemma D.1.** *(**Lemma 4** in [31]) Let $v \in \mathbb{R}^{|\mathcal{S}|}$ and $0 < q < \Delta(\mathcal{S})$. The value of optimization problem*

$$\min_{p \in \Delta(\mathcal{S})} \langle p, v \rangle \text{ such that } KL[p||q] < C_{KL} \tag{35}$$

*is equal to*

$$\min_{\theta \geq 0} \theta C_{KL} + \theta \ln \left( \langle q, \exp\left(-\frac{v}{\theta}\right) \rangle \right). \tag{36}$$

*Let $\theta^*$ be the solution of equation (36), then the solution of (35) becomes,*

$$p \propto q \exp\left(-\frac{v}{\theta^*}\right). \tag{37}$$

Using lemma D.1, Equation (34) can be implemented as

$$Q_{c_n}^{(t+1)}(s,a) = c_n(s,a) + \gamma \sum_{s \in \mathcal{S}} P_{(s,a)}^*(s') V_{c_n}^{(t)}(s'),$$

$$\text{where } P_{(s,a)}^* \propto p^0(.|s,a) \exp\left( \frac{V_{c_n}^t(.)}{\theta_{(s,a)}^*} \right), \tag{38}$$

$$\text{and } \theta_{(s,a)}^* := \arg\min_{\theta \geq 0} \theta C_{KL} + \theta \ln \left( \langle p^0(.|s,a), \exp\left( \frac{V_{c_n}^t(.)}{\theta} \right) \rangle \right).$$

---

**Algorithm 3** Robust_Q-table

---

1: **Input:** $c_i, \pi, p^0, C_{KL}, \rho$
2: **Initialize:** $Q(s,a) = 0 \; \forall(s,a) \in \mathcal{S} \times \mathcal{A}, V(s) = 0 \; \forall s \in \mathcal{S}, Q_{prev}(s,a) = 0 \; \forall(s,a) \in \mathcal{S} \times \mathcal{A}$
3: $s = \rho(.), \; \tau = 1000, \; i = 1$
4: **while** $i < \tau$ **do**
5: $\quad Q_{prev}(s,a) = Q(s,a) \; \forall(s,a) \in \mathcal{S} \times \mathcal{A}$
6: $\quad a = \pi(.|s)$
7: $\quad s' = p^0(.|s,a)$
8: $\quad P^* = \dfrac{p^0[s,a,.] \exp\left(\frac{V[.]}{C_{KL}}\right)}{\sum_{s' \in \mathcal{S}} p^0[s,a,s'] \exp\left(\frac{V[s']}{C_{KL}}\right)} \; \forall(s,a)$
9: $\quad Q[s,a] = c_i[s,a] + \gamma\langle P^*, V\rangle$
10: $\quad V[s] = \langle\pi[.|s], Q(s,.)\rangle \; \forall s \in \mathcal{S}$
11: $\quad s = s'$
12: $\quad$ **if** $Q(s,a) = Q_{prev}(s,a) \; \forall(s,a) \in \mathcal{S} \times \mathcal{A}$ **then**
13: $\quad\quad$ Break out of loop
14: $\quad$ **end if**
15: $\quad i = i + 1$
16: **end while**
17: **Return:** $Q, V$

---

While Equation (36) is convex in nature, solving it for all $p(.|s,a)\forall(s,a) \in (\mathcal{S}, \mathcal{A})$ in Equation (38) is computationally extensive in practice. Rather than the exact constrained problem, [58] proposed a **regularized robust DP update**.

$$Q_{c_n}^{(t+1)}(s,a) = c_n(s,a) + \gamma \max_{p \in \Delta_{\mathcal{S}}} \left( \sum_{s' \in \mathcal{S}} p(s') V_{c_n}^t(s') - C_{KL}' KL[p || p^0(.|s,a)] \right), \quad (39)$$

where $C_{KL}' > 0$ is a constant. This regularized form can be efficiently written as Equation (40)

$$Q_{c_n}^{(t+1)} = c_n(s,a) + \gamma \left( \sum_{s' \in \mathcal{S}} P_{(s,a)}^*(s') V_{c_n}^t(s') \right),$$

$$\text{where } P_{(s,a)}^* \propto p^0(.|s,a) \exp\left(\frac{V_{c_n}^t(.)}{C_{KL}'}\right). \quad (40)$$

The equivalence can be concluded from the duality since it is convex optimization problem. The following lemma also shows that the convergence is fast.

**Lemma D.2.** *(Adaptation from **Proposition 3.1** and **Theorem 3.1** [58]) For any $C_{KL}' > 0$, there exists $C_{KL} > 0$ such that Equation* (39) *converges linearly to the fixed point of Equation* (38).

# E Experiments

The environments where we test our algorithms are as given below (Some results are shown in the main paper under Experiments section (Section 5)). Before moving on to the individual environment, we first state the hyper-parameters that are fixed throughout the environments.

**Common hyperparameters**

The initial state distribution, denoted by $\rho$, is generated by sampling from a standard normal distribution followed by applying a softmax transformation to convert the resulting values into a valid probability distribution over states. In particular, for each state, a random number is generated from $\mathcal{N}(0,1)$. Then it is normalized using softmax in order to avoid negative values.

The discount factor $\gamma$ is set to 0.99 across all algorithms and environments to ensure consistency. However, in order to evaluate computational efficiency (wall-clock time), we run EPIRC_PGS with multiple discount factors: $\gamma = 0.9, 0.99,$ and $0.995$.

`EPIRC_PGS` follows a double-loop structure, as described in [8], where the outer loop uses the iteration index $K$ and the inner loop uses index $T$. In our experiments, we set $K = 10$ and $T = 100$, yielding a total of $K \times T = 1000$ iterations. This ensures that all algorithms are compared over the same number of update steps.

Both `RPPG` and `RNPG` require an initial policy specification. For `RPPG`, we initialize the policy uniformly: $\pi^0(a \mid s) = 1/|\mathcal{A}|$ for all $s \in \mathcal{S}$. In contrast, `RNPG` parameterizes the policy directly using a vector $\theta$, where $\theta^0 \sim \mathcal{N}(0, 1)$ and $|\theta^0| = |\mathcal{S}| \times |\mathcal{A}|$.

Both algorithms also depend on the hyperparameter $\lambda$. For `RNPG`, $\lambda$ is fixed at 50 across all experiments. For `RPPG`, $\lambda$ is treated as a variable hyperparameter, with values specified individually in the corresponding experimental sections.

The learning rate $\alpha$ is set to $10^{-3}$ for all algorithms across all environments. Another important hyperparameter is the loop control variable $\tau$, used in Algorithm 3. The operations inside the loop of Algorithm 3 represent a robust Bellman update. It has been shown in [31] that the soft Bellman operator is a contraction mapping. Therefore, setting $\tau$ to a large value ensures convergence to a fixed point $Q(s, a)$, and subsequently to the corresponding value function $V(s)$. In our experiments, we fix $\tau = 1000$. For theoretical justification, refer to Lemmas D.1 and D.2

## E.1 Constrained River-swim

The River-swim environment is a widely studied benchmark in optimization theory and stochastic control. The detailed explanation of the algorithm is as given below.

### E.1.1 Environment Description

The environment consists of six distinct states, conceptualized as islands dispersed across a large body of water. At the start of each episode, a swimmer is placed on one of these landmasses.

The swimmer's objective is to navigate toward one of the two terminal islands—representing the river's endpoints—to receive a reward. At each state, the swimmer can choose between two actions: swimming to the left or to the right. Rewards are only provided upon reaching the terminal states, whereas all intermediate states yield zero reward (refer to Table 2).

During the transition between states, the swimmer encounters adversarial elements, such as strong water currents and hostile tribal inhabitants residing on certain islands. These hazards are modeled as a cost incurred for occupying a given state. The transition probabilities between states are compactly represented in Table 2, while the immediate state-wise rewards and constraint costs are summarized in Table 3. Note that the reward is high at the extreme right-hand side as this is the best state, however, it also corresponds to high current or high cost. All the parameters including the value of $C_{KL}$ of the MDP are represented in Table 4.

| State | Action | Probability for next state |
|---|---|---|
| $s_0$ | $a_0$ | $s_0$:0.9, $s_1$:0.1 |
| $s_i, \quad i \in \{1, 2, \ldots 5\}$ | $a_0$ | $s_i$:0.6, $s_{i-1} : 0.3$, $s_{i+1} : 0.1$ |
| $s_i, \quad i \in \{0, 1, \ldots 4\}$ | $a_1$ | $s_i$:0.6, $s_{i-1} : 0.1$, $s_{i+1} : 0.3$ |
| $s_5$ | $a_1$ | $s_5$:0.9, $s_4$:0.1 |

Table 2: Transition probability of River-swim environment

| State | Reward | Constraint cost |
|---|---|---|
| $s_0$ | 0.001 | 0.2 |
| $s_1$ | 0 | 0.035 |
| $s_2$ | 0 | 0 |
| $s_3$ | 0 | 0.01 |
| $s_4$ | 0.1 | 0.08 |
| $s_5$ | 1 | 0.9 |

Table 3: The reward and constraint cost received at each state

| | Hyperparameters | Value |
|---|---|---|
| Environment Parameters | $|\mathcal{S}|$ | 6 |
| | $|\mathcal{A}|$ | 2 |
| | $p^0$ | Table 2 |
| | $c_0, c_1$ | Table 3 |
| | $b$ | 42 |
| KL Uncertainity Evaluator (Algorithm 2) | $\gamma$ | 0.99 |
| | $\alpha_{kle}$ | $10^{-4}$ |
| | $C_{KL}$ | 0.1 |
| Robust_Q_table | $\alpha$ | $10^{-4}$ |
| RPPG | $\lambda$ | 10 |

Table 4: Hyperparameter used for all subroutines for CRS environment

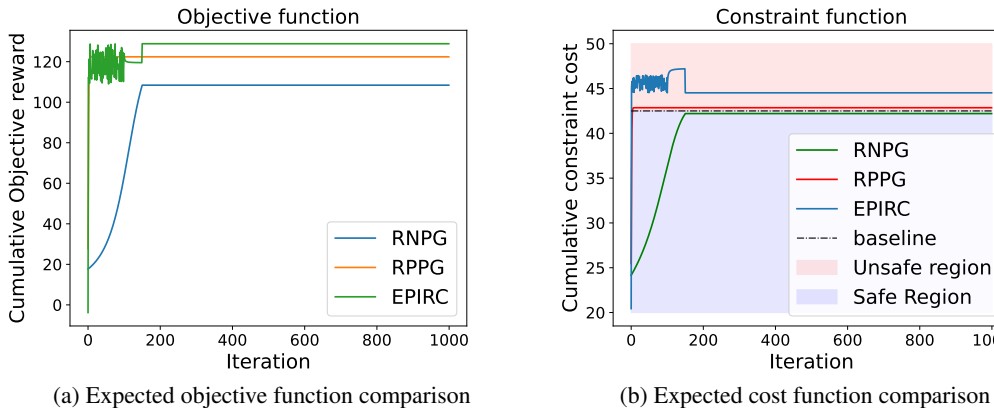

(a) Expected objective function comparison  (b) Expected cost function comparison

Figure 3: Comparison of RPPG and EPIRC-PGS on CRS environment

### E.1.2 Discussions of the result

The iteration-wise expected reward (value function) and expected constraint cost are illustrated in Figure 3. From Figure 3a, we observe that EPIRC_PGS (denoted as EPIRC) achieves the highest objective reward. *However, as shown in Figure 3b, it significantly violates the constraint threshold, failing to remain within the designated safe region.* Since the agent's goal is not only to maximize long-term reward but also to ensure safety by satisfying the constraint, EPIRC_PGS falls short in this regard.

RPPG achieves a higher value function than RNPG, as seen in Figure 3a. However, a closer look at Figure 3b reveals that RPPG also marginally violates the constraint boundaries. RNPG effectively captures the trade-off between reward maximization and the constraint satisfaction, navigating as close as possible to the constraint boundary. It stops at the point where further increase in reward would result in constraint violations, thereby maintaining a feasible and safe policy.

Our algorithm relies on a key hyperparameter, $\lambda$. This parameter plays a crucial role in balancing the objective and constraint terms during policy updates. Specifically, $\lambda$ should be chosen to be sufficiently large such that when the constraint violation $J_{c_i}^{\pi_t} - b_i$ is marginal (i.e., $J_{c_i}^{\pi_t} - b_i > \xi$ for some small $\xi > 0$ and for any $i \in 1, 2, \ldots, K$), the scaled objective term $J_{c_0}^{\pi_t}/\lambda$ does not dominate the update direction.

If $\lambda$ is set too small, the influence of the objective term becomes large. As a result, the algorithm may prioritize minimizing the objective cost (or maximizing the reward, depending on the environment setting) at the expense of constraint satisfaction. This contradicts our goal of maximizing the expected objective return such that the expected constraint values are below a certain threshold. To illustrate the impact of $\lambda$ on the performance and feasibility of RNPG, we conduct experiments using different values of $\lambda$, with results presented in Figure 4. Note that higher value of $\lambda$ indeed reduces the value function, but also decreases the cumulative cost. We set $\lambda = 50$ throughout the

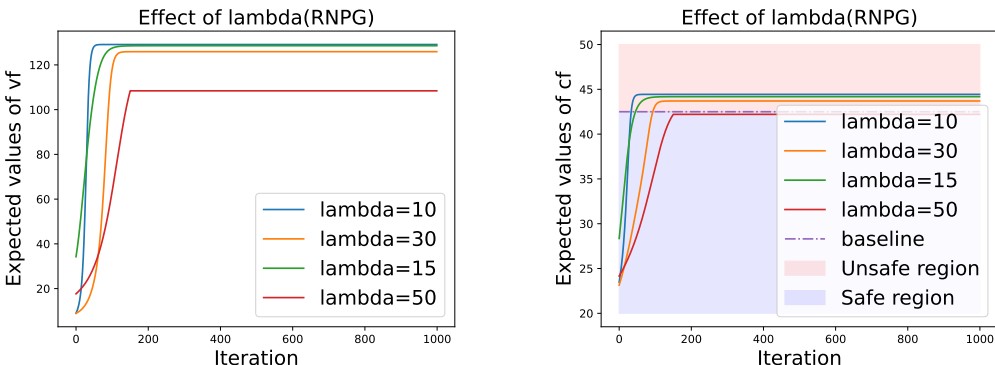

Figure 4: Effect of $\lambda$ on RNPG for the CRS environment

experiment for RNPG as it corresponds to feasible solution for each environment. Hence, it shows that for RNPG, we do not need to costly hyper-parameter tuning for $\lambda$ as a relatively high value of $\lambda$ ensures feasibility as the Theory suggested.

Furthermore, Table 11 presents a comparison of wall-clock time across the algorithms. RNPG completes in the shortest time, running approximately $1.6\times$ faster than RPPG and at least $4\times$ faster than EPIRC_PGS (at $\gamma = 0.9$). These results demonstrate that RNPG not only achieves competitive performance but also does so with significantly improved computational efficiency compared to both RPPG and EPIRC_PGS.

*The results highlight RNPG's ability to consistently learn robust and safe policies while outperforming RPPG and EPIRC-PGS in terms of both reliability and computational efficiency, even under adverse environmental dynamics.*

### E.2 Garnet problem

#### E.2.1 Environment Description

The *Garnet environment* is a standard Markov Decision Process (MDP) framework commonly used to evaluate reinforcement learning (RL) algorithms in a controlled setting. It is characterized by a predefined number of states $nS$ and actions $nA$, where the transition probabilities, rewards, and utility functions are randomly sampled from specified distributions. The transition dynamics in Garnet are typically sparse, meaning that each state does not transition to all other states, but instead has a limited number of possible successor states for each action. Mathematically, the environment is defined by a transition probability matrix $P(s' \mid s, a)$, a reward function $R(s, a)$, and, in the case of constrained RL, a utility function $U(s, a)$. These elements are often drawn from normal distributions, i.e.,

$$P(s' \mid s, a) \sim \mathcal{N}(\mu_a, \sigma_a), \quad R(s, a) \sim \mathcal{N}(\mu_b, \sigma_b), \quad U(s, a) \sim \mathcal{N}(\mu_c, \sigma_c)$$

.

where the means $\mu_a, \mu_b, \mu_c$ are sampled from a uniform distribution $\text{Unif}(0, 100)$. Since the transition probability matrix must be valid (i.e., each row should sum to 1), the probabilities are exponentiated and normalized using a softmax transformation:

$$p^0(s' \mid s, a) = \frac{\exp(P(s' \mid s, a))}{\sum_{s''} \exp(P(s'' \mid s, a))}$$

.

#### E.2.2 Implementation details

In this environment, both the reward and cost values are stochastic, sampled randomly rather than being deterministically assigned. The cost (or reward) values are generated as follows:

$$R = c_0 \sim \mathcal{N}(\mu_b, \sigma_b). \tag{41}$$

where $\mu_b \sim \mathcal{U}[0, 10]$ and $\sigma_b = 1$. Similarly we generate the cost function. Here, we use $c_0$ and $R$ interchangeably because the Garnet environment is formulated as a reward-based MDP with a utility-based constraint function. Unlike the Constrained River-swim environment, the objective here is to maximize the long-term expected reward while ensuring that the expected utility remains above a specified threshold.

The hyperparameters used for this environment are listed in Table 5

| | **Hyperparameters** | **Value** |
|---|---|---|
| Environment Parameters | $|\mathcal{S}|$ | 15 |
| | $|\mathcal{A}|$ | 20 |
| | $b$ | 90 |
| KL Uncertainity Evaluator (Algorithm 2) | $\gamma$ | 0.99 |
| | $\alpha_{kle}$ | $10^{-3}$ |
| | $C_{KL}$ | 0.05 |
| Robust_Q_table | $\alpha$ | $10^{-3}$ |
| RPPG | $\lambda$ | 15 |

Table 5: Hyperparameter used for all subroutines for Garnet environment

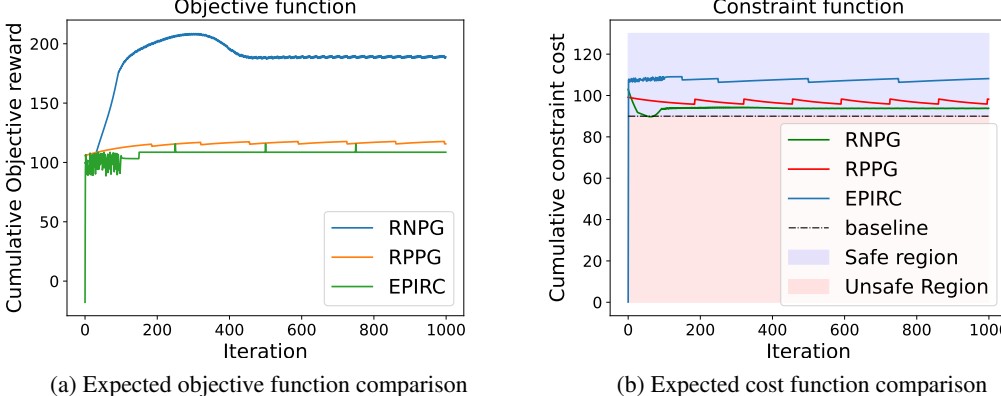

(a) Expected objective function comparison

(b) Expected cost function comparison

Figure 5: Comparison of RPPG and EPIRC-PGS on Garnet(15,20) environment $\lambda = 30$

### E.2.3 Discussion of Results

The results are shown in Figure 5. As previously discussed, the Garnet environment incorporates a utility function in the constraint rather than a traditional cost function. Therefore, a feasible optimal policy is expected to yield an expected utility (constraint) value that remains above a predefined threshold. For consistency in terminology and to avoid confusion, we refer to the utility function as the "constraint function" in Figure 5b.

From Figure 5b, it is evident that all three algorithms—RNPG, RPPG, and EPIRC_PGS—satisfy the constraint throughout training, thus producing feasible policies at each iteration. However, Figure 5a shows that RNPG achieves a noticeably higher expected objective return compared to both EPIRC_PGS and RPPG.

Figure 5 provides further insight into RNPG's behavior. Initially, RNPG operates well within the safe region and progressively improves its objective return. As it approaches the constraint boundary, the algorithm detects the potential violation and adjusts its trajectory accordingly—prioritizing safety over additional reward. This contrasts with the behavior of RPPG and EPIRC_PGS, which also maintain feasibility but yield comparatively lower objective returns. These results highlight the advantage of incorporating a natural policy gradient approach, which allows RNPG to balance safety and performance more effectively.

In addition to performance, we compare the computational efficiency of the algorithms. Table 11 shows that RNPG requires a computation time comparable to RPPG, but significantly outperforms EPIRC_PGS in terms of speed. Specifically, RNPG is at least $5\times$ faster than EPIRC_PGS when $\gamma = 0.9$, and nearly $8\times$ faster when $\gamma = 0.995$. The increased runtime for EPIRC_PGS at higher discount factors is attributed to the longer binary search required for constraint satisfaction as $\gamma$ approaches 1. *The key takeaway from this experiment is that RNPG demonstrates greater sensitivity to constraint boundaries and exhibits strong potential for scalability to larger state and action spaces. Notably, the Garnet environment used in this study contains 15 states and 20 actions. These results suggest that, with efficient implementation, RNPG can be effectively extended to high-dimensional settings.*

### E.3 Modified Frozen-lake

The general Frozen-lake is as special type of grid world problem. The vanilla Frozen-lake problem can be found in gymnasium library [59]. However, in this work, we create a small modification to make the problem more challenging and interesting.

#### E.3.1 Environment description

The Frozen Lake environment is modeled as a $d \times d$ grid world, where the agent begins its journey at the top-left corner, $s_0 = (0, 0)$, and aims to reach the bottom-right goal state $s_{d^2-1} = (d-1, d-1)$. At each time step, the agent may choose one of four primitive actions: move *left*, *right*, *up*, or *down*, constrained by the grid boundaries.

The environment contains multiple types of states:

- **Goal state:** Reaching the terminal state yields a *high reward*.
- **Hole states:** If the agent steps into a hole, it *falls in* and receives a *very low reward*.
- **Normal states:** All other transitions yield a *moderate reward*.

In addition to reward dynamics, the environment contains *hazardous blocks*, which are selected randomly at each iteration. These represent dynamic threats (e.g., thin ice, traps, or roaming predators) and impose a *high constraint cost* when visited. The stochastic nature of these threats introduces uncertainty in the agent's experience, making the problem both risky and difficult to optimize.

The agent's objective is to learn a policy that maximizes the expected cumulative reward while incurring only marginal harm. In other words, it must learn to reach the goal while *minimizing the cumulative constraint cost* associated with hazardous states.

We formulate this problem as a *Constrained Markov Decision Process (CMDP)* under model uncertainty. For all experiments, we set the grid size to $d = 4$. To map a 2D coordinate $(x, y)$ to its corresponding 1D state index, we define a wrapping function:

$$\text{wrap}((x, y)) = x \times d + y.$$

The probability distribution function is shown in Table 6.

The rewards and cost functions are given in Table 7. In particular, if it reaches the goal state the reward is $+1$. If the agent hits the obstacle, the cost is 1, and and the frozen grid it is $0.3$. Note that the a grid is obstacle or not is decided randomly at the beginning of an episode.

We detail all the other parameters in Table 8.

#### E.3.2 Discussion of results

The results obtained of the given environment is depicted in Figure 6. As seen in Figure 6b, all the three algorithms successfully learns the feasible policies. However, on observing Figure 6a, we can clearly notice the dominance of RNPG by learning policies with better rewards. From Table 11, we see that for the frozen lake environment, the computation time of RPPG and RNPG is almost comparable. However, RNPG is atleast 3x as faster as EPIRC-PGS for $\gamma = 0.9$ and almost 4x faster than EPIRC-PGS for $\gamma = 0.995$.

*The key takeaway from this enviornment is to show that even with added obstacles randomly, the agent can find a feasible high objective return policy as compared to RPPG and EPIRC-PGS*

| Present state | action | Transition probabilities |
|---|---|---|
| $(x=0,y=0)$ | up | $(x=0,y=0):2/3,(x+1,y)=1/6$ and $(x,y+1)=1/6$ |
| $(x=0,y\neq0)$ | up | $(x=0,y):1/2,(x+1,y)=1/6,(x,y+1)=1/6$ and $(x,y-1)=1/6$ |
| $(x\neq0,y\neq0)$ | up | $(x-1,y):1/2,(x+1,y)=1/6,(x,y+1)=1/6$ and $(x,y-1)=1/6$ |
| $(x=0,y=0)$ | left | $(x=0,y=0):2/3,(x+1,y)=1/6$ and $(x,y+1)=1/6$ |
| $(x\neq0,y=0)$ | left | $(x,y=0):1/2,(x+1,y)=1/6,(x-1,y)=1/6$ and $(x,y+1)=1/6$ |
| $(x\neq0,y\neq0)$ | left | $(x,y-1):1/2,(x+1,y)=1/6,(x,y+1)=1/6$ and $(x-1,y)=1/6$ |
| $(x=3,y=3)$ | down | $(x=3,y=3):2/3,(x-1,y)=1/6$ and $(x,y-1)=1/6$ |
| $(x=3,y\neq3)$ | down | $(x=3,y):1/2,(x-1,y)=1/6,(x,y+1)=1/6$ and $(x,y-1)=1/6$ |
| $(x\neq3,y\neq3)$ | down | $(x+1,y):1/2,(x-1,y)=1/6,(x,y+1)=1/6$ and $(x,y-1)=1/6$ |
| $(x=3,y=3)$ | right | $(x=3,y=3):2/3,(x-1,y)=1/6$ and $(x,y-1)=1/6$ |
| $(x\neq3,y=3)$ | right | $(x,y=3):1/2,(x+1,y)=1/6,(x-1,y)=1/6$ and $(x,y-1)=1/6$ |
| $(x\neq3,y\neq3)$ | right | $(x,y-1):1/2,(x+1,y)=1/6,(x,y+1)=1/6$ and $(x-1,y)=1/6$ |

Table 6: Transition probabilities for Frozen lake environement

| State | Reward | constraint cost |
|---|---|---|
| (x=3,y=3) | +1 | 0 |
| (x,y) if frozen lake | 0 | 0.3 |
| (x,y) if obstacle | 0.01 | 1 |
| all other (x,y) | 0.05 | 0 |

Table 7: State wise rewards and constraint cost for the Frozen lake environment

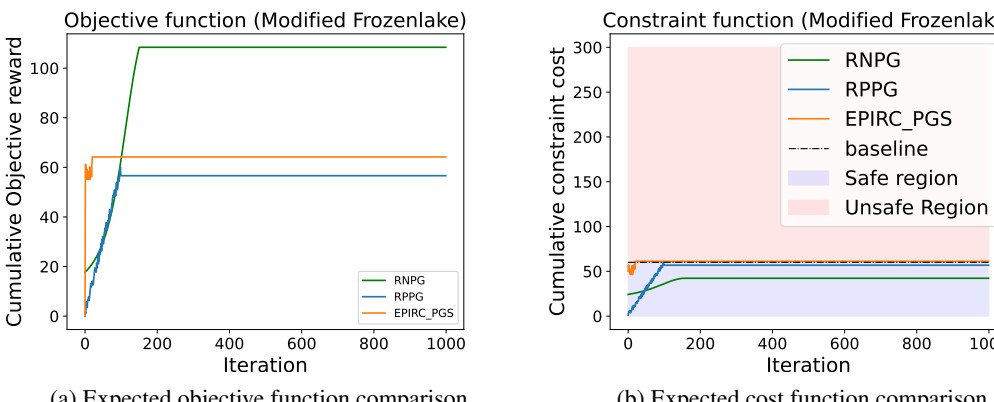

(a) Expected objective function comparison (b) Expected cost function comparison

Figure 6: Comparison of RNPG, RPPG and EPIRC-PGS on Modified Frozen-lake environment

## E.4 Garbage collection problem

### E.4.1 Environment description

We model a city as a $4 \times 4$ grid, where each cell represents a city block. A garbage collection robot is deployed to navigate this grid and collect waste while minimizing operational risk and resource expenditure.

Certain blocks offer higher rewards due to significant waste accumulation (e.g., near hospitals or markets). However, urban conditions are inherently dynamic. These high-reward blocks are not known in advance and are constantly changing showing the rapid changes of city areas. At each time step, $40\%$ of the blocks are randomly designated as *hazardous*, representing unpredictable real-world events such as:

- Sudden traffic congestion
- Unreported toxic waste dumps
- Temporary road closures or civil disturbances

| | Hyperparameters | Value |
|---|---|---|
| Environment Parameters | $\|\mathcal{S}\|$ | 15 |
| | $\|\mathcal{A}\|$ | 4 |
| | $p^0$ | Table 6 |
| | $c_0, c_1$ | Table 7 |
| | $b$ | 55 |
| KL Uncertainity Evaluator (Algorithm 2) | $\gamma$ | 0.99 |
| | $\alpha_{kle}$ | $10^{-3}$ |
| Robust_Q_table | $C_{KL}$ | 0.02 |
| | $\alpha$ | $10^{-3}$ |
| RPPG | $\lambda$ | 50 |

Table 8: Hyperparameter used for all subroutines for Modified Frozen-lake environment

These hazardous blocks incur a higher constraint cost if visited. Importantly, the set of hazardous blocks changes **at every iteration**, introducing a layer of real-time uncertainty in the environment.

The robot must learn a policy that balances the dual objectives of:

1. Maximizing long-term reward by collecting from high-value blocks
2. Minimizing cumulative constraint costs induced by environmental hazards

The transition probabilities are similar to the Frozenlake environment. Hence the transition probabilities for this environment can be depicted by Table 6. The reward and cost structure is given in Table 9. While the reward is fixed at the Goal location, the reward at garbage location is 0.01. Note that whether a certain grid is a garbage location or not is decided randomly. Similarly, the cost is 1 at a blocked grid. Again, the identities of the blocked grids are randomly decided.

| State | Reward | Cost incurred |
|---|---|---|
| (x=3,y=3) | 1 | 0.01 |
| (x,y) if garbage | 0.01 | - |
| (x,y) if blockage | - | 1 |
| (x,y) all other state | 0.001 | 0.01 |

Table 9: Reward and cost structure for Garbage collector environment

The hyperparameters for the various sub-routines are as listed in Table 10.

| | Hyperparameters | Value |
|---|---|---|
| Environment Parameters | $\|\mathcal{S}\|$ | 15 |
| | $\|\mathcal{A}\|$ | 4 |
| | $p^0$ | Table 6 |
| | $c_0, c_1$ | Table 9 |
| | $b$ | 60 |
| KL Uncertainity Evaluator (Algorithm 2) | $\gamma$ | 0.99 |
| | $\alpha_{kle}$ | $10^{-3}$ |
| Robust_Q_table | $C_{KL}$ | 0.02 |
| | $\alpha$ | $10^{-3}$ |
| RPPG | $\lambda$ | 50 |

Table 10: Hyperparameter used for all subroutines for Garbage collector environment

### E.4.2 Discussion of results

In this subsection, we will present the performance of the RPPG, EPIRC_PGS and our algorithm (RNPG)(Figure 7). As shown in Figure 7b, due to the randomness of the environment, the algorithms have some minor fluctuations. However, in this environment, RPPG and RNPG obey the constraints for the complete duration, but, EPIRC_PGS violates the constraint. Although none of the algorithms

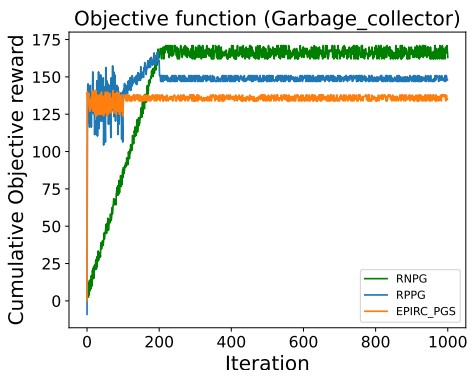
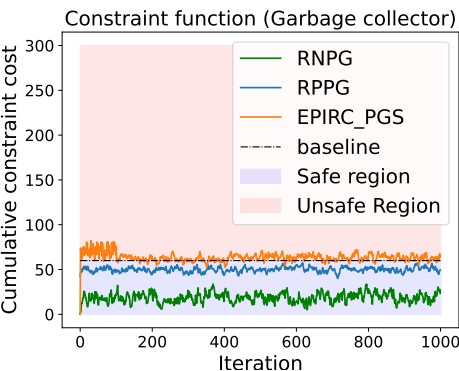

(a) Expected objective function comparison  (b) Expected cost function comparison

Figure 7: Comparison of RNPG, RPPG and EPIRC-PGS (or, EPIRC_PGS) on Garbage collector environment

| Best policy's objective and constraint return comparison | | | | | | | | | | |
|---|---|---|---|---|---|---|---|---|---|---|
| **Env Nm** | **RPPG** | | **RNPG** | | **EPIRC-PGS** ($\gamma$ vals.) | | | | | |
| | | | | | 0.9 | | 0.99 | | 0.995 | |
| | $v_f$ | $c_f$ | $v_f$ | $c_f$ | $v_f$ | $c_f$ | $v_f$ | $c_f$ | $v_f$ | $c_f$ |
| **CRS,** $b_1 = 42.5$ | 120.3 | 43.1 | 102.1 | 42.2 | 127.2 | 44.8 | 121.2 | 48.1 | 123.4 | 46.4 |
| **Garnet,** $b_1 = -90$ | 102.6 | -96.5 | 208.1 | -98.7 | 100.3 | -112.4 | 100.6 | -108.3 | 102.4 | -110.2 |
| **MFL,** $b_1 = 52.5$ | 61.4 | 53.1 | 109.2 | 39.2 | 62.3 | 53.7 | 60.3 | 57.7 | 66.3 | 51.7 |
| **GC,** $b_1 = 52.5$ | 161.2 | 49.2 | 172.2 | 22.1 | 131.2 | 52.0 | 142.5 | 55.6 | 138.1 | 54.1 |

Table 11: Comparison of the best policy objective ($v_f$) and constraint function ($c_f$) values. $b_1$ indicates the threshold value. RNPG not only achieves the best value, but also gives a feasible policy.

stabilize completely yet RPPG and RNPG are in the safe zone. In terms of objective return, it can be seen from Figure 7a, the expected return for RNPG is predominantly higher than RPPG and EPIRC_PGS. While comparing the time (Table 11) for completion RPPG is the fastest in this environment marginally beating RNPG but still the speeds of both algorithms are comparable. When compared with EPIRC_PGS, RNPG is winning fairly with a speedup of 2x compared to EPIRC_PGS when $\gamma = 0.9$ and a speedup of nearly 3x compared to EPIRC_PGS when $\gamma = 0.995$.

*This environment demonstrates that, even under random obstacle placement, the agent can successfully learn a feasible policy that outperforms RPPG and EPIRC_PGS in terms of objective return.*

## F  Implementation Details of RNPG and RPPG

### F.1  RNPG

Note that in (11) one can use direct parameterization for policy update in RNPG. To facilitate optimization, we also adopt a soft-max representation of the policy space. Let the policy be parameterized by $\theta$, such that:

$$\pi^{\theta_t}(a \mid s) = \frac{\exp\left(\theta_t[s]\right)}{\sum_{s \in \mathcal{S}} \exp\left(\theta[s]\right)}. \tag{42}$$

Using this parameterization, we reformulate the policy update as the solution to the following constrained optimization problem:

$$\theta_{t+1} \in \arg\max_{\theta_{t+1}} \left\langle \nabla J_{c_{\text{ch}}}^{\pi_{\theta_t}}, \theta_{t+1} - \theta_t \right\rangle - \alpha_t \, \text{KL}(\pi_{\theta_{t+1}} \| \pi_{\theta_t}), \tag{43}$$

where the objective index ch is selected as:

$$\text{ch} = \arg\max \left\{ \frac{J_{c_0}^{\pi_{\theta_t}}}{\lambda}, \max_{i=1,\dots,K} \left( J_{c_i}^{\pi_{\theta_t}} - b_i \right) \right\}.$$

This formulation enables us to apply the Natural Policy Gradient method by incorporating the geometry of the policy space through the Fisher Information Matrix $\mathcal{F}$ [60]. The resulting closed-form update rule is:

$$\theta_{t+1} = \theta_t - \alpha_{\text{lr}} \cdot \frac{1}{2\alpha_t} \mathcal{F}^{-1} \nabla J_{c_{\text{ch}}}^{\pi_{\theta_t}}.$$

## F.2 Robust Projected Policy Gradient (RPPG)

We also compare the Robust Projected Policy Gradient (RPPG) which uses $\ell_2$ regularization instead of the KL regularization. Here, we use direct parameterization. The policy update is given in the following.

$$\pi_{t+1} \in \arg\min_{\pi \in \Pi} \langle \nabla_{\pi_t} J_i(\pi_t), \pi - \pi_t \rangle + \frac{1}{2\alpha_t} \|\pi - \pi_t\|^2, \tag{44}$$

where $i = \arg\max \left\{ \frac{J_{c_0}^{\pi}}{\lambda}, \max_n \left( J_{c_n}^{\pi} - b_n + \xi \right) \right\}$.

Upon careful observation, we see that Equation (44) is convex. To find the optimal solution of $\pi_{t+1}$, we use projected gradient descent. Equation (44) can be updated as

$$\pi_{t+1} = \arg\min_{\pi \in \Pi} \left\| \pi - (\pi_t - \alpha_t \nabla_{\pi_t} J_i^{\pi_t}) \right\|^2. \tag{45}$$

This is the Euclidean projection of the gradient step onto the simplex:

$$\pi_{t+1} = \Pi_\Delta \left( \pi_t - \alpha_t \nabla_{\pi_t} J_i(\pi_t) \right) \tag{46}$$

From Lemma 3.1, we get the value of $\nabla_{\pi_t} J_i(\pi_t)$ using the robust Q value evaluator in Algorithm 3. We finally project the resulting value into the policy space simplex, $\Pi$. To perform projection, we find $\|\pi - (\pi_t - \alpha_t \nabla_{\pi_t} J_i(\pi_t))\|_2 \ \forall \ \pi \in \Pi$. However, this process is cumbersome, hence we can leverage the cvxpy package from Python to optimally solve the update equation.

---

**Algorithm 4** Robust-Projected Policy Gradient for CMDP with uncertainties (R-PPG)

---

1: **Input:** Robust Policy evaluator (Algorithm 2), $b_i \quad s.t. \ i \in \{1, K\}, \xi, \lambda$
2: **Initialization**: $\hat{\pi}(\cdot|s)_0 = 1/|A|$ for all $s$, T.
3: **for** $t = 0, \ldots, T-1$ **do**
4:      Evaluate $J_j^{\pi_t} = \max_P J_{j,P}^{\pi_t}$ for $j = \{c_0 \ldots c_K\}$ using the robust policy evaluator oracle.
5:      $ch = \arg\max(J_{c_0}^{\pi_t}/\lambda, J_{c_i}^{\pi_t} - b_i + \xi) \ s.t. \ i \in \{1, K\}$
6:      **if** $ch \neq 0$ **then**
7:          $\pi_{t+1} = \text{Proj}_\Pi \{\pi_t - \alpha_t \nabla J_{c_{ch}}^{\pi_t}\}$.
8:      **else**
9:          $\pi_{t+1} = \text{Proj}_\Pi \{\pi_t - \alpha_t \nabla J_{c_0}^{\pi_t}/\lambda\}$
10:      **end if**
11: **end for**
12: Output $\hat{\pi} = \arg\min_{t \in 0,\ldots,T-1} \max\{J_{c_0}^{\pi_t}/\lambda, \max_i\{J_{c_i}^{\pi_t} - b + \xi\}\}$

---

As shown in Algorithm 4, RPPG leverages Projected Policy Gradient method to reach the optimal policy. In general $\ell_2$ regularizer does not ensure small changes in the policies and might deviate a lot from the previous policy. Thus, KL-regularizer has a better performance over $\ell_2$ regularizer which we further demonstrate by our results in Section E.

## G    Extension to Continuous state space (Robust Constrained Actor Critic)

We present our robust constrained actor–critic framework designed for the function approximation setting as discussed in Section 6.1. The model comprises two critic networks—one estimating the reward value function $J_{c_0}$ and the other corresponding to the Constraint value function $J_{c_1}$. Although we focus on a single constraint for clarity, the framework readily generalizes to handle multiple constraints. In addition to the critic networks, an actor network is employed to generate actions based on the current state. To model distributional robustness, we consider IPM as described in Section 6.1.

Let us consider the critic network be parameterized by $w$ where each layer contains $d$ paramters including the bias (i.e, $w_{1:d}^l$ where $w_1^l$ is assumed to be the bias term in the $l$-th layer with $l \in \{1, 2, \ldots, L\}$). Overall approach is depicted in Algorithm 5.

---

**Algorithm 5** Robust Constrained Actor Critic (RCAC)

---

1: **Input:** $T, \rho, b$
2: **Initialization**: $w_r$ (for objective estimation), $w_c$ (for constraint estimation) and $\theta^0$ for actor network parameterization
3: **for** $t = 0, \ldots, T - 1$ **do**
4:      Get estimate for $J_r = \langle \rho, V_{w_r}(s) \rangle$ and $J_c = \langle \rho, V_{w_c}(s) \rangle$
5:      $ch = \arg \max(J_r/\lambda, (J_c - b))$
6:      Update $w_{ch}$ using $w_{ch} = w_{ch} + \alpha_k.\nabla_{w_{ch}} \text{MSE}(\langle \rho, V_t(s) \rangle, J_{ch})$ (Note $V_t(s)$ is target Value function obtained by Robust_TD_update (using equation (15)))
7:      Update $\theta$ using $\theta^t = \theta^{t-1} + \alpha.\mathbb{E}\left[\nabla_\theta \log(\pi_\theta(a|s)).(Q_{ch}(s, a) - V_{ch}(s))\right]$ *(We change this step to Natural Policy Gradient update for RCAC_NPG).*
8: **end for**

---

At each step, we get the Value function estimate $V_r$ and $V_c$ from the respective Critic networks. After obtaining both, we make a choice as to whether to update the constraint critic parameters or the objective critic parameters. For the selected critic, we find the target value function using the robust bellman operator along with a guided regularization term on the last layer only [54]. We compute the robust value according to (15).

For our experiments, we chose the famous Cartpole environment, where the intial state is fixed and deterministic so $\rho(i)$ is a unit vector.(However, it can be extended to different distribution.)

$$\rho(s) = \begin{cases} 1 & \text{if } s = s_0 \\ 0 & \text{otherwise} \end{cases} \tag{47}$$

In our study, we introduced uncertainty in the next-state transition after each action. While alternative sources of uncertainty could be incorporated—such as perturbing the executed actions or simulating external disturbances (e.g., wind forces acting on the cart)—we focused on state transition perturbations because they have a more direct and analyzable impact on value estimation. Perturbing the action space was deemed less meaningful in this environment, as the action set is discrete with only two possible values, making the resulting learning challenge comparatively trivial. The detailed results and observations are presented in the following subsection.

### G.1 Results and discussion

In this sub-section, we list the results obtained when we tried our algorithm against the standard cartpole-v1 environment available in gymnasium library. The cartpole-v1 algorithm comprises of a continuous state space having 4 components and two discrete actions. We introduce uncertainity by adding noise to the next state obtained after taking an action. The noise is adding a uniform value between 0 and 0.1 to the original next state value ($s' = s' + Unif([0, 0.1])$) and then clipped it between the predefined bounds of cartpole environment.

We divided the experiments results into two phases. The first is the training phase (depicted by Figures 8) and the second is during the testing phase (depicted by Figures 9). During the training phase, we only train the robust variants RCAC, RCAC with NPG, Robust CRPO, EPIRC-PGS by considering $\delta = 0.04$. However, for constrained actor-critic (CAC), we do not train the robust version. During the training phase (Figure 8), apart from EPIRC-PGS, all the algorithms perform similarly in terms of reward and the cost value function (We highlight our two algorithms RCAC with NPG, and RCÅC, separately in figure 10). EPIRC-PGS did not converge and could not complete the entire episode highlighting that binary search approach is not possible to scale for large state-space. However, when these algorithms were tested (Figure 9) on the environment having a perturbation uniformly between 0 and 0.04, the performance of CAC is unstable and did not provide any feasible policy. Only, our proposed approaches achieve feasibility while being close to the optimality. Robust CRPO also violates the constraint (slightly) while achieving less reward compared to our approaches. The performances of the algorithms is compactly represented in table 12.

| Algorithm | Average reward value | Average cost value |
|---|---|---|
| Constrained Actor Critic | 495.23 | 294.6248 |
| **RCAC** | **494.81** | **177.8432** |
| **RCAC with NPG** | **495.0** | **197.9937** |
| Robust CRPO | 488.06930 | 213.473 |
| EPIRC_PGS | 114.8 | 53.79 |

Table 12: Tabular comparisons of the average value function and cost function during the testing phase. CAC although returns policies with high objective function but the actions are unsafe as can be inferred from the high constraint function (safety baseline is 200)

It is also important to note down the wall clock time for the various algorithms. EPIRC_PGS takes the highest wall clock time approximately 24029.013 seconds which is nearly $4\times$ of the time taken for the other algorithms namely *RCAC with NPG* (**7175.31 seconds**), *CAC*(**6815.89 seconds**), *RCAC* (**5975.65 seconds**) and *Robust CRPO* (**4275.67 seconds**) in decreasing order of the wall clock time requirements. 9.

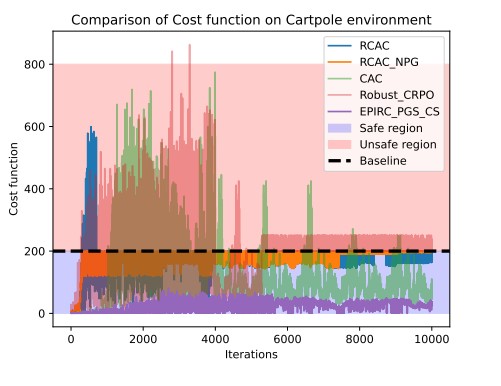
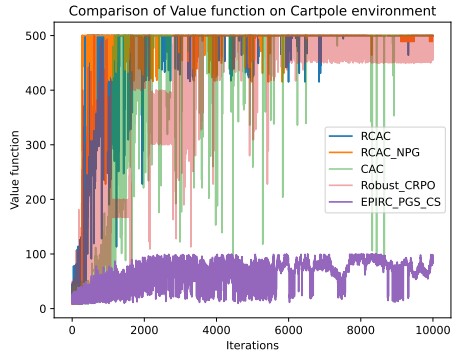

(a) Value function comparison        (b) Cost function comparison

Figure 8: Comparison of RCAC, RCAC_NPG, robust CRPO and other standard Constrained MDP solutions on the Cartpole problem during training phase

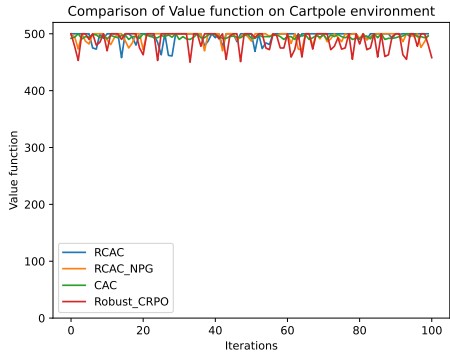
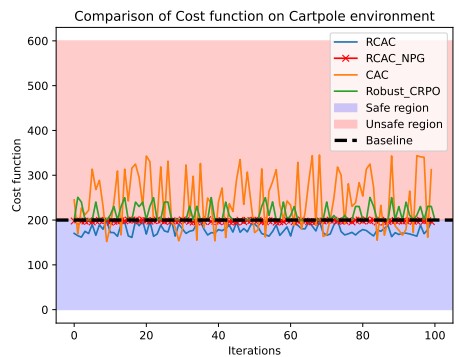

(a) Value function comparison        (b) Cost function comparison

Figure 9: Comparison between RCAC, CRPO and Vanilla Constrained Actor Critic during testing period $\delta = 0.04$ (deflection from the nominal model)

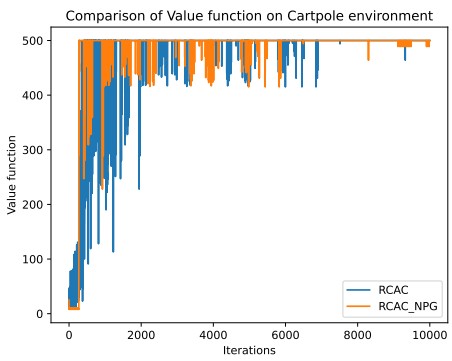
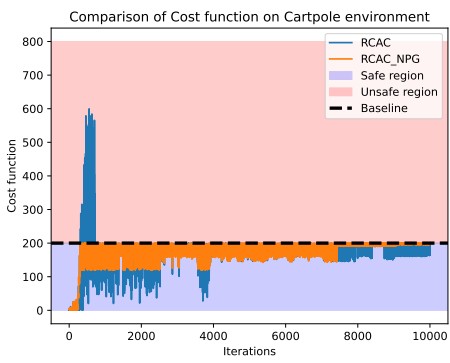

(a) Value function comparison
(b) Cost function comparison

Figure 10: The comparison plots between our two main variants of Robust Constrained Actor Critic variants (RCAC and RCAC_NPG)

# H  Connection with the CRPO

CRPO is one of the popular approach for non-robust CMDP which has been proposed in [11]. In the CRPO algorithm, one minimizes the objective when all the constraints are satisfied, and minimizes the constraint value if the policy violates the constraint for at least one constraint. In particular, the objective function can be represented as

$$\min_{\pi} J_{c_0}^{\pi} \mathbb{1}(\max_n J_{c_n} - b_n \leq 0) + \max_n (J_{c_n} - b_n) \mathbb{1}(\max_n J_{c_n} - b_n > 0). \tag{48}$$

Thus, one might think that there are some connections with our approach and the robust CRPO. First, we provide the challenges in extending the results to the RCMDP case. In [11], they bound the difference in the value function corresponding to the policies between two steps using the standard value difference lemma. However, the standard value difference lemma does not hold in the robust case as the worst case transition probabilities differ according to the probabilities.

In what follows, we point out the difference of our approach and potentially robust CRPO approach. In order to obtain iteration complexity, we seek to use the smoothness property of the objective by invoking Moreu's envelope as done in [8]. In particular, we use a smooth function $\max\{J_r^{\pi}/\lambda, \max_n J_{c_n}^{\pi} - b_n\}$ as an objective instead of the one in (48). It turns out the this modification is essential for obtaining the iteration complexity. Note the difference–we are not switching to minimize the constraint cost value functions when the constraints are not satisfied, rather we are only minimizing those when $\max_n J_{c_n}^{\pi} - b_n$ becomes larger than $J_{c_0}^{\pi}/\lambda$. Thus, as $\lambda$ becomes larger it becomes similar to CRPO. Also, note that in the asymptotic sense as $\lambda \to \infty$, we can not guarantee the sub-optimality gap anymore showing that perhaps, robust CRPO algorithm may not achieve the iteration complexity bound.

