# OpenReview forum: "Efficient Policy Optimization in Robust Constrained MDPs with Iteration Complexity Guarantees"
_NeurIPS.cc/2025/Conference — NeurIPS 2025 poster_

### Official Review · Reviewer_XfhM · 2025-06-28

**Clarity:** 3
**Significance:** 2
**Originality:** 3
**Rating:** 4
**Confidence:** 3

**Summary:**

This paper introduces a novel approach designed for the Robust CMDP (RCMDP) problems. The authors first reformulate the optimization problem in a new way to balance the trade-off between optimizing the objective and satisfying the constraints, and then propose an algorithm PNPG which is at most $\epsilon$ sub-optimal. This paper mainly has two contributions: one is that there is no need to perform a binary search on $\lambda$, compared to the state-of-the-art method, and the other is that it first shows the strict feasibility can be achieved and improves the iteration complexity to $O(\xi^{-2}\epsilon^{-2})$ if the strict feasibility parameter $\xi$ is known. The authors further illustrate that if $\xi$ is unknown, the algorithm can get a policy with $O(\epsilon^{-4})$ iteration complexity, which has a same dependence on $\epsilon$ but a significantly better dependence on $S, A, (1-\gamma)^{-1}$ compared to the state-of-the-art method. Finally, the authors conduct numerical experiment for their algorithm and the experiment settings and results make sense to me.

**Questions:**

1. Since the paper considers policy optimization, it is kind of weird to restrict the results to only the tabular case where the state space and action space is finite. Is it easy to extend to infinite state-action space setting?

2. Is there any lower bound on the iteration complexity?

3. It seems that the reference [10] belongs to the LP-based methods for solving the primal problem directly, instead of addressing the iteration complexity?

**Ethical Concerns:**

["NO or VERY MINOR ethics concerns only"]

**Final Justification:**

I maintain my positive recommendation of the paper.

**Limitations:**

yes.

**Paper Formatting Concerns:**

No.

**Quality:**

3

**Strengths And Weaknesses:**

Strength: The paper is well-written and it is an easy read for the reader to get the main idea. Also, in contrast to previous works on robust CMDP problem (EPIRC-PGS), this paper presents a computationally more efficient approach (RNPG) without binary search, achieving a faster iteration complexity bound and better dependence on the state-space. Compared to RPPG, the approach (RNPG) has better performance and more stability.

Weakness 1: The paper does not consider infinite state-space which may be very different from finite state.

Weakness 2: The authors do not prove the optimality for their iteration complexity lower bound. There may exist some better approaches to achieve tighter results.

---

> ### Author Rebuttal · Authors · 2025-07-28
>
> We appreciate your time and thoughtful evaluation of our paper. We recap your comment and present our detailed response as follows. We would also be happy to provide further clarifications if suitable.
>
> >*Since the paper considers policy optimization, it is kind of weird to restrict the results to only the tabular case where the state space and action space is finite. Is it easy to extend to infinite state-action space setting?*
>
> * We now report results in **continuous (infinite) state spaces** using an actor–critic variant on CartPole. Specifically, we extend our method to a Robust Constrained Actor–Critic (RCAC) framework in which a neural‑network critic approximates Integral Probability Metric (IPM)‑based worst‑case values  [A1] and the actor is updated via natural policy gradient. Training RCAC on the standard CartPole benchmark with IPM ambiguity set demonstrates that our approach scales beyond tabular settings to continuous state spaces with function approximation.
>
> | Algo | Avg. Reward Value | Avg. Cost-value | Wall-Clock Time (s) |
> |---|---|---|---|
> |Vanilla Constrained Actor-Critic with NPG | 323.15  | 350.12 | 752.35|
> |**RCAC** | **298.99** | **199.2**| 770.25 |
> | CRPO | 352.42 | 370.69 |761.56  |
> |Robust CRPO | 281.24 | 179.7 | 701.1|
> |EPIRC-PGS|  114.8 | 53.79 | 1429.14|
>
> For the CartPole environment, we use the perturbation of $\delta=0.1$. We considered the cost function to be the absolute distance of the cart position from the center ($0$). Thus, in other words, the constraint function ensures that the cart remains as close to the centre as possible.
> The threshold for the constraint is 200. **From the table, it is evident that among all the feasible approaches, our approach (RCAC) achieves the best reward.** In particular, it significantly outperforms the EPIRC-PGS both in terms of reward and the wall-clock time. The main issue with EPIRC-PGS appears to be that the binary search is prone to error, particularly in the function-approximation setup, as the worst-case model might not be exact.
>
>
>
> * **Limitation**. A complete theory for general function approximation lies beyond the scope of this paper. That said, the structure of IPM ambiguity sets yields closed‑form worst‑case value updates under linear function classes and a tractable surrogate in continuous (infinite) state spaces, suggesting a viable path to extend our analysis. Importantly, this extension is nontrivial: it requires (i) uniform deviation bounds for the IPM‑robust Bellman operator over rich function classes (e.g., covering‑number/Rademacher analyses), (ii) a careful bias–variance/error‑propagation analysis for an approximate worst‑case critic and its effect on natural policy‑gradient updates, (iii) the impact on the point-wise maximum over different value functions in the surrogate objective, and (iv)  iteration‑complexity bounds that account for model misspecification. Our new experiments indicate that the method behaves well in practice; developing these guarantees is a substantive avenue for future work.
>
>
> >*Is there any lower bound on the iteration complexity?*
>
> * The lower bound of iteration complexity, even for unconstrained robust MDP, remains open. Characterizing the lower bound indeed or achieving a better upper bound for RCMDP  constitutes an important future research question.
>
> >*It seems that the reference [10] belongs to the LP-based methods for solving the primal problem directly, instead of addressing the iteration complexity?*
>
> * The reviewer is correct. We will specify that.
>
> [A1]. Zhou, Ruida, et al. "Natural actor-critic for robust reinforcement learning with function approximation." Advances in neural information processing systems 36 (2023): 97-133.

---

> > ### Author Response · Authors · 2025-08-06
> >
> > Dear Reviewer XfhM,
> >
> > We would like to thank you for your time and thoughtful evaluation of our paper. We are glad that you positively evaluate our contributions. Since the author-reviewer discussion period is ending soon, and we have not heard from you, we just wanted to check in and ask if the rebuttal clarified and answered the questions raised in your review. We would be very happy to engage further if there are additional questions!

---

> > > ### Comment · Reviewer_XfhM · 2025-08-08
> > >
> > > Yes, my concerns are addressed well by the authors and I will maintain my already positive recommendation of the paper and consider raising the score in the final recommendation.

---

> > > > ### Author Response · Authors · 2025-08-08
> > > >
> > > > We are glad that our responses have resolved your questions. We thank you for your positive recommendation and consideration of raising the score further.

---

### Official Review · Reviewer_rBgq · 2025-06-29

**Clarity:** 4
**Significance:** 3
**Originality:** 4
**Rating:** 4
**Confidence:** 4

**Summary:**

The authors develop an algorithm for performing reinforcement learning in a constrained Markov decision process (MDP) setting. In regular constrained MDPs, the state-action occupancy measures of the underlying policies are convex, and the min-max problem can be solved via strong duality. On the other hand, if the underlying transition function of the MDPs is not known, the problem is no longer convex and is hard to solve. The authors develop a projected gradient method like existing constrained MDP algorithms, but rely on a KL-divergence regularization term to provide stability. Specifically, the algorithm first maximizes the reward like regular RL algorithms, but the algorithm adjusts the KL-divergence term if a constraint is violated during the training to provide safety. The algorithm is evaluated on two toy examples, and the authors show that the robust natural policy gradient (RNPG) algorithm can provide superior performance while satisfying the safety constraints.

**Questions:**

1- The authors assume that the initial policy is strictly feasible, which may not be the case in many scenarios. The authors also mention that this assumption (Assumption 1) can be relaxed in Theorem 6.1 by setting \lambda = 2H/ \epsilon and the proof is easy to follow afterward. The authors mentioned that the algorithm would have an O(\epsilon^{-4}) in this case, similar to the algorithm in reference [8]. I was curious if the authors could comment on the performance of the RNPG algorithm and [8] if the algorithms are initialized with an infeasible policy.

2- The authors mentioned that Assumption 2 is a strict one as the Assumption assumes that the transition functions of each set have a similar probability (based on \Beta) for each successive states, which allows the authors to use Lemma B.3 in the Appendix for calculating an upper bound of difference in the policy update. I do agree that this assumption is restrictive, particularly on the transition functions sharing the same support. I was curious if the authors could comment on replacing that Assumption with a weaker term such as having a bound on the expected successive states in a discrete or continuous state-action setting. I'd think that such an assumption would be more applicable in a continuous state-action setting rather than a discrete one.

3- The authors used a KL-regularizer term for stability in their proofs. I was curious if the authors could comment on other regularization functions or formulations for ensuring robust feasibility. For example, I was curious if the authors could make a comparison with the set-based reachability analysis compared to using a KL-divergence term for regularization.

**Ethical Concerns:**

["NO or VERY MINOR ethics concerns only"]

**Final Justification:**

I did go through the responses of the authors to my and other reviews. I think the authors did a good job in clarifying the required assumptions, and also the added experiments to my and Reviewer Vtph's comments. Therefore, I am raising my score.

**Limitations:**

Yes.

**Paper Formatting Concerns:**

N/A.

**Quality:**

3

**Strengths And Weaknesses:**

The strengths of the paper are providing an efficient algorithm for the robust contained MDP problem by using KL-divergence term instead of projected gradient ascent in the policy update to provide superior computation efficiency compared to existing work [8]. The algorithm improves on the existing iteration complexity O(\epsilon^{4}) to O(\epsilon^{2}).

However, the authors assume that the initial policy is strictly feasible. They discuss how this assumption can be relaxed in Section 6, and provide a short proof and bound in the Appendix. However, the authors assume that the transition functions of the MDP has the same support, i.e., given a state and action, they should have the same set of successive states, which I believe is a bigger restriction compared to the finite state and action spaces.

---

> ### Author Rebuttal · Authors · 2025-07-28
>
> We appreciate your time and thoughtful evaluation of our paper. We recap your comment and present our detailed response as follows. We would also be happy to provide further clarifications if suitable.
>
> >*The authors assume that the initial policy is strictly feasible, which may not be the case in many scenarios.The authors also mention that this assumption (Assumption 1) can be relaxed in Theorem 6.1 by setting \lambda = 2H/ \epsilon and the proof is easy to follow afterward. The authors mentioned that the algorithm would have an O(\epsilon^{-4}) in this case, similar to the algorithm in reference [8]. I was curious if the authors could comment on the performance of the RNPG algorithm and [8] if the algorithms are initialized with an infeasible policy.*
>
> *  Thank you for pointing this out. The reviewer’s interpretation is not accurate:
>
> * **No strict‑feasibility requirement for the starting policy.** Our algorithm can be initialized with an infeasible policy. For all our experiments, the initial policy is a random policy that can be both in the feasible and infeasible region. We had further experiments where the initial policies were strictly infeasible, yet the algorithms were able to quickly move to safer regions despite initially starting from the unsafe regions.
>   * We ran our algorithms on the Garnet environment and the Garbage Collector environment. For the Garnet environment, our algorithm started with a policy that returned an objective value function (maximize it) of 103.12 and has a cost value function of 98.5 (which is infeasible as it needs to be above $100$).  However, the final policy has an objective value function and constraint value function of 175.11 and 120.00, respectively.
>
>   * For the Garbage collector, the Initial value function (maximize it) is 220.216, with an initial cost function (which should lie below 60) of 82.02. The final value function and constraint cost function are 170.21 and 56.15, respectively.
>
> * **Clarification of Assumption 1.** We assume only that the *optimal policy* is strictly feasible. Under this condition, the algorithm produces a policy that is atmost $\epsilon$ sub-optimal and feasible within $O(\epsilon^{-2})$ iterations. No constraint is imposed on the feasibility of the initial policy.
>
> We will revise the manuscript to emphasize this distinction and prevent further confusion.
>
> >*The authors mentioned that Assumption 2 is a strict one as the Assumption assumes that the transition functions of each set have a similar probability (based on \Beta) for each successive states, which allows the authors to use Lemma B.3 in the Appendix..*
>
> * Thank you for raising this point. Relaxing **Assumption 2** is indeed a key direction for future work. As the reviewer correctly notes, if the perturbed distribution assigns positive probability to an event, the nominal model should also assign positive probability to that event. *Otherwise, a mismatch in supports could lead to unsampled regions and render finite‑sample bounds intractable, hence, it seems that this assumption is essential.* *While this assumption is essential for establishing our theoretical guarantees, we deliberately did not enforce it in the empirical study;* the algorithm still performed well, suggesting that the practical impact may be less restrictive than the theory implies. A rigorous treatment of differing supports—and corresponding error bounds—remains an open and important challenge. Also, note that a similar assumption is also made for unconstrained robust MDP [A1]. Further, **EPIRC-PGS also assumed that the ratio between the state-action occupancy measures on the states covered by all policies and the initial state distribution is bounded.** Hence, it seems that one needs an assumption similar to Assumption 2 to achieve theoretical guarantees for robust MDP and RCMDP.
>
> >*The authors used a KL-regularizer term for stability in their proofs. I was curious if the authors could comment on other regularization functions or formulations for ensuring robust feasibility. For example, I was curious if the authors could make a comparison with the set-based reachability analysis compared to using a KL-divergence term for regularization.*
>
> * Our algorithm can work with other uncertainty sets except the KL-uncertainty sets, where there are effective approaches to find the worst-case value function. For example, we have now achieved results for continuous state-space with the Integral probability metric (IPM), where we used $L_2$-norm as a regularization to train the critic network.  We extend the work to a Robust Constrained Actor Critic (RCAC) paradigm, which uses Neural networks to approximate the critic and uses natural policy gradient to update the actor network. We train the RCAC algorithm on the standard Cartpole environment with the Integral Probability Metric (IPM) as the ambiguity set for the worst-case value function. The result shows that our work can be substantially scaled to a continuous state space.
>
> | Algo | Avg. Reward Value | Avg. Cost-value | Wall-Clock Time (s) |
> |---|---|---|---|
> |Vanilla Constrained Actor-Critic with NPG | 323.15  | 350.12 | 752.35|
> |**RCAC** | **298.99** | **199.2**| 770.25 |
> | CRPO | 352.42 | 370.69 |761.56  |
> |Robust CRPO | 281.24 | 179.7 | 701.1|
> |EPIRC-PGS|  114.8 | 53.79 | 1429.14|
>
> For the CartPole environment, we use the perturbation of $\delta=0.1$. We considered the cost function to be the absolute distance of the cart position from the center ($0$). Thus, in other words, the constraint function ensures that the cart remains as close to the centre as possible.
> The threshold for the constraint is 200. **From the table, it is evident that among all the feasible approaches, our approach (RCAC) achieves the best reward.** In particular, it **significantly** outperforms the EPIRC-PGS both in terms of reward and the wall-clock time.
>
>
> [A1]. Zhou, Ruida, et al. "Natural actor-critic for robust reinforcement learning with function approximation." Advances in neural information processing systems 36 (2023): 97-133.

---

> > ### Comment · Reviewer_rBgq · 2025-08-04
> >
> > I appreciate the detailed response to my questions and reviews. Specifically, I think the clarifications about assumptions and KL-divergence regularization and the subsequent extension (RCAC) look more promising. Therefore, I am raising my score to borderline accept.

---

> > > ### Author Response · Authors · 2025-08-04
> > >
> > > We would like to thank the reviewer for raising the score and appreciating our contributions. We will add the discussions and the new results on RCAC to the final version.

---

### Official Review · Reviewer_Vtph · 2025-06-29

**Clarity:** 1
**Significance:** 2
**Originality:** 3
**Rating:** 4
**Confidence:** 5

**Summary:**

This paper considers the setting of robust constrained MDPs, which aims to minimize the expected cumulative cost for the cost function $c_0$ and hope maintain a feasible cost for the cost function $c_i$ for $i=1,2,...,K$. An existing approach applies the epigraph approach to solve this problem but it requires to evaluate robust value function at every iteration and has $O(\epsilon^{-4})$ iteration complexity. This paper aims to address this issue with proposing a more computationally efficient approach. The proposed method re-formulates the problem design with re-writing the objective loss in an equivalent form and this paper derives the best-known iteration complexity. The results are also validated using experiments.

**Questions:**

I have a few requests on improving this manuscript:
1. Add empirical experiments on mojuco or other more common RL environments.
2. Fully revise the proof to improve the readability and the correctness.
3. Add experiments to validate the compatibility with neural network and considering the continuous state or action spaces.

**Ethical Concerns:**

["NO or VERY MINOR ethics concerns only"]

**Final Justification:**

I have carefully checked the revised comment provided by the author. Given the proposed fix, the proof should be correct without major alter from its original result. I still encourage the author to fully revise the proof to improve the readability.

I am not completely satisfied with the experiments, but the newly added results have been partially sufficient to support the results presented in this paper.

As the result, I raise my score to 4.

**Quality:**

2

**Strengths And Weaknesses:**

## Strengths
This paper proposes a new approach to handle the robust constrained RL problem. The re-formulated objective loss is new and has not been discussed before. To this reason, this paper is sufficiently new. Proving the equivalence between the re-formulated objective (3) and the original objective (1) also introduces a new theoretical result.

## Weaknesses
1. Only toy experiments are provided. It is not irrational to claim that the algorithm is more computationally efficient with providing only toy examples.
2. The given Garnet and Constrained River-swim are too trivial.  The proposed method needs to be compared with more methods such as [1] in more tasks (e.g., mujoco locomotion tasks) to demonstrate its efficiency.

[1] Xu, Tengyu, Yingbin Liang, and Guanghui Lan. "Crpo: A new approach for safe reinforcement learning with convergence guarantee." International Conference on Machine Learning. PMLR, 2021.

3. After a careful check of the proof, I found several critical issues that cannot be simply fixed during the rivision period:
    * The whole proof doesn't specify the optimized parameter in the function, especially when using $\max$ or $\arg\max$. For example, in L599, it doesn't tell what the p is. It could be taken over P, and it can also be taken over $\pi$ (p for Pi). The context is unclear so the subscript shouldn't be omitted.
    *  Many statements are incomplete. For example, Lemma B.3. only presents an equation without introducing the notations used in this equation.
    *  Due to the unclear notations and incomplete statements, many steps in the proof are confusing and hard to check if it is correct or not:
        * In Equation (30), the first inequality and the second inequality are the same. No explanation given here on why it holds.
        * Equation (31) is not clear. The explaination "$V(s') \geq V^{\pi^\*}(s')$ as $J-J^{\pi^\*}\geq 0$" doesn't hold. Because $J$ is an expected $V^\pi$, the non-negative expected value doesn't imply each atom is non-negative.
        * After Equation (31): the recursively expanding step omitts too much. First, the RHS of Lemma B.3 is missing one $\rangle$ immediately after $\pi^*$. Second, the RHS looks so different from Equation (31). More details are needed.
I didn't further check the remaining proof as the manuscript should be fully revised to improve its readablity and correctness. The current version is obviously flawed.

4. This paper relies on a very specific uncertainty set. Though it is claimed that "Our framework is not limited to KL-divergence", the explaination and the empirical analysis are not sufficiently convincing. Also, it requires to take the specific policy evaluation algorithm used in existing literature, which makes this paper lack sufficient novelty in this direction.

5. The proposed methods cannot be extended to the general function approximation. The author claims this in the limitation section but it doesn't prevent it from being a major limiation of this  proposed method.

---

> ### Author Rebuttal · Authors · 2025-07-28
>
> We appreciate your time and thoughtful evaluation of our paper. We recap your comment and present our detailed response as follows. We would also be happy to provide further clarifications if suitable.
>
> >*Only toy experiments are provided. It is not irrational to claim that the algorithm is more computationally efficient by providing only toy examples...*
>
> * We appreciate the concern. Our claim is about **relative computational efficiency**—*for a fixed robust‑evaluator*—and follows from RNPG’s algorithmic structure.  Methods such as EPIRC‑PGS require an **outer search/binary-search** over the epigraph parameter, which can be significant when the worst-case value function times increase (e.g., TV/Wasserstein) or for infinite state-space where the worst-case evaluator can be noisy. In particular, EPIRC-PGS needs to find the maximum policy for each parameter $b_0$, and then needs to tune that parameter using a binary-search method; our approach does not require that. Hence, **it is quite evident that our approach is efficient**.
>
> ---
>
> ### Evidence beyond small/tabular settings
>
> **Function approximation & continuous state space.**
>    We add a **Robust Constrained Actor–Critic (RCAC)** with a neural critic (IPM‑based worst‑case estimator) and **natural policy‑gradient** actor updates. On **CartPole (continuous state)** with perturbations, RNPG attains **feasibility and high return** using **mini‑batch** updates and an **approximate** robust oracle—demonstrating that benefits are not artifacts of tabularity.
>
> | Algo | Avg. Reward Value | Avg. Cost-value | Wall-Clock Time (s) |
> |---|---|---|---|
> |Vanilla Constrained Actor-Critic with NPG | 323.15  | 350.12 | 752.35|
> |**RCAC** | **298.99** | **199.2**|770.25|
> | CRPO | 352.42 | 370.69 |761.56  |
> |Robust CRPO | 281.24 | 179.7 | 701.1|
> |EPIRC-PGS|  114.8 | 53.79 | 1429.14|
>
> For the CartPole environment, we use the perturbation of $\delta=0.1$. The threshold for the constraint is 200. **From the table, it is evident that among all the feasible approaches, our approach achieves the best reward.** In particular, it significantly outperforms the EPIRC-PGS both in terms of reward and the wall-clock time.
>
> 2.**Strength against EPIRC-PGS**.
>
> Also, we would urge the reviewer to see the results on  **sparse‑transition** domains (e.g., RiverSwim variants), larger $\gamma$-values,  and larger tabular MDPs with increasing state counts. As the worst-case evaluation time increases, our approach achieves a better result compared to EPIRC-PGS.
>
> We would also like to bring to the reviewer’s attention that our **supplementary** paper also lists a few experiments motivated from the gymnasium or safety-gymnasium library specially designed for CMDP studies, such as the Modified frozen-lake and the Garbage collector where we consistently outperforms EPIRC-PGS.
>
> >*Regarding Proof of Lemma B.3..*
>
> * We apologize for an oversight. First, there are typos in the proof of Lemma B.3. In particular, $\pi^*$,
>
>  should be replaced by $\hat{\pi}^*$ , the optimal solution of the surrogate problem in (9). As Lemma B.3 only deals with the surrogate problem.
>
> * Second, we will explicitly mention that $\hat{\pi}^*$
>
> is a uniform minimizer of the surrogate problem (we have removed $\lambda$ to maintain notational simplicity for Markdown)
>
> $\min_{\pi}\max_{i}[\max_PV^{\pi,P}_{c_i}(s)] \forall s$.
>
> We will update eq.(9) with this modification. **Note that, we do not need this assumption on the $\pi^*$ of the original RCMDP. Rather, we need it for the optimal solution for the surrogate problem in (9).** Note that this is similar to the Assumption made for the unconstrained setup as well [A1,A2].
>
> * Third, using this, we can now expand in the following way, for a given $s$, $\max_{j}(\max_{p}V_{c_j}^{\pi,p}(s))-\max_{j}(\max_{p}V_{c_j}^{\hat{\pi}^*,p}(s))$
>
> $ \leq V_{c_i}^{\pi,p}(s)$
> $-V_{c_i}^{\hat{\pi}^*, p}(s)$,
>
>
> where we use the fact that $(p,c_i)=\arg\max_{j,p} V_{c_j}^{\pi,p}(s)$
>
>
> * Fourth, we also missed $p$ in the last line of (31), in particular $Q^{\pi}$, and $V_{\pi}$ will be associated with $p$. (Also, the notation $\pi_s$, simply means $\pi(\cdot|s)$; we have used this notation in (11) as well, perhaps we will redefine it again to avoid confusion).
> Hence, in eqn (31), we apply the fact that
>
> $V_{c_i}^{\pi,p}(s)=\max_{j,p}(V_{c_j}^{\pi,p}(s))\geq \max_{j,p}(V_{c_j}^{\hat{\pi}^*,p}(s))$
>
> $\geq V_{c_i}^{\hat{\pi}^*,p}(s)$
>
> because of the uniform minimizer assumption on  $\hat{\pi}^*$ $\forall s$, this allows us to use Assumption 2, and use  $p_0$
>
> in place of  $p$  . We then expand the second term again and complete the proof by Induction. Lemma B.3 mimics Lemma 9 in [A1] for the RCMDP.  Since it is true for every atom, we can then easily show that for any initial state distribution $\rho$, concluding it for $J$ which we need to show for Lemma 4.2
>
> >*I didn't further check the remaining proof as the manuscript should be fully revised to improve its readablity and correctness.*
>
> * **We believe that we have taken utmost care to avoid any proof bugs.** We respectfully ask the reviewer to reexamine the revised proofs; if any particular step remains unclear, we will gladly provide an even more detailed, line referenced justification in the supplement.
>
> * To summarize, the only concern of the reviewer was the proof of Lemma B.3 because there were some typos (minor issues) and the lack of justification on $V_{c_i}^{\pi,p}(s)\geq V_{c_i}^{\hat{\pi}^*,p}$.  Lemma B.3 only concerns the surrogate optimization problem, not the original optimization problem. We have now fixed the issue on the assumption of the uniform maximizer of the surrogate problem in (9) by $\min_{\pi}\max_{i}[\max_PV^{\pi,P}_{c_i}(s)] \forall s$. **Crucially, this assumption is only on the surrogate problem, not on the original problem, and an Assumption that is also used in the unconstrained version [A1,A2].** **Further, this fix is local and does not alter any theorems, rates, or conclusions: the statements and proofs go through verbatim after the correction, and all empirical results are unchanged.** We have also strengthened our paper with new results for large state-space (CartPole).
>
> >*This paper relies on a very specific uncertainty set. Though it is claimed that "Our framework is not limited to KL-divergence", the explanation and the empirical analysis are not sufficiently convincing.*
>
> * As we mentioned, our contribution is not in terms of developing a policy evaluation algorithm; rather, our contribution is towards **provable iteration complexity guarantee for RCMDP**. It is not trivial, as, **unlike the CMDP, we cannot use the traditional primal-dual-based approach because the worst-case value function is different for the objective and the constraint,** rendering the robust dynamic programming approach inapplicable on the composite value function. We therefore introduce a **surrogate optimization problem, prove its tight relationship to the original RCMDP,** and design an algorithm that achieves provable iteration complexity for the surrogate despite the additional pointwise‑max operator that is absent in the unconstrained case. In contrast to EPIRC‑PGS, **our method avoids the epigraph/binary‑search loop, leading to faster convergence in practice.** Further, since the binary search loop requires an accurate evaluation of the worst-case value function, EPIRC-PGS is not suitable for a large-scale setup. While our theory is developed in the tabular/linear regime, the formulation and algorithm extend to continuous (infinite) state spaces with function approximation, unlike the EPIRC-PGS—supported by new experiments—leaving a full general‑FA theory as important future work.
>
> >*The proposed methods cannot be extended to the general function approximation...*
>
> *  While our theory does not yet cover general function approximation, the methodology itself extends naturally. In the revision, we add an actor–critic instantiation of RNPG, robust constrained actor-critic (RCAC) for large state spaces: the critic is trained as an IPM‑based worst‑case value estimator (approximate robust oracle) [A1] from samples, and the actor is updated with RNPG using mini‑batches. On the perturbed CartPole, this implementation attains high returns with feasibility, indicating that RNPG is practically applicable beyond the tabular setting. The comparison with the other approaches is shown below:
>
> | Algo | Avg. Reward Value | Avg. Cost-value | Wall-Clock Time(s) |
> |---|---|---|---|
> |Vanilla Constrained Actor-Critic with NPG | 323.15  | 350.12 | 752.35|
> |**RCAC** | **298.99** | **199.2**|770.25 |
> | CRPO | 352.42 | 370.69 |761.56  |
> |Robust CRPO | 281.24 | 179.7 | 701.1|
> |EPIRC-PGS|  114.8 | 53.79 | 1429.14|
>
> For the CartPole environment, we use the perturbation of $\delta=0.1$. The threshold for the constraint is 200. We considered the cost function to be the absolute distance of the cart position from the center ($0$). Thus, in other words, the constraint function ensures that the cart remains as close to the centre as possible.  **From the table, it is evident that among all the approaches that are feasible, our approach achieves the best reward.** In particular, it **significantly** outperforms the EPIRC-PGS both in terms of reward and the wall-clock time.
>
>
> **In short, although a complete general‑FA proof is outside the present scope, our new experiments demonstrate that RNPG does extend in practice to function approximation and large state spaces, mitigating the reviewer’s concern.**
>
> [A1]. Zhou, Ruida, et al. "Natural actor-critic for robust reinforcement learning with function approximation." Advances in neural information processing systems 36 (2023): 97-133.
>
> [A2]. Iyengar, Garud N. "Robust dynamic programming." Mathematics of Operations Research 30, no. 2 (2005): 257-280.

---

> > ### Comment · Reviewer_Vtph · 2025-08-03
> >
> > I have carefully checked the revised comment provided by the author. Given the proposed fix, the proof should be correct without major alter from its original result. I still encourage the author to fully revise the proof to improve the readability.
> >
> > I am not completely satisfied with the experiments, but the newly added results have been partially sufficient to support the results presented in this paper.
> >
> > As the result, I raise my score to 4.

---

> > > ### Author Response · Authors · 2025-08-03
> > >
> > > We would like to thank the reviewer for increasing their score and appreciating our contributions. We will definitely add the modified version to improve readability. We will also incorporate the new empirical results (and additional large-scale experimentations) in the final version.

---

### Official Review · Reviewer_GbJz · 2025-07-02

**Clarity:** 3
**Significance:** 3
**Originality:** 3
**Rating:** 5
**Confidence:** 3

**Summary:**

The paper introduces a KL‑regularized policy‑gradient method (RNPG) that compare favorably to EPIRC‑PGS in terms of computational efficiency and proves an $O(\epsilon^{-2})$ convergence rate to zero‑violation solutions in robust CMDPs.

**Questions:**

- How would the results change if working with a finite state-action space but with function approximation?
- It might have been interesting to also see a baseline of a naive Lagrange method or something like CRPO applied with worst-case values.

**Ethical Concerns:**

["NO or VERY MINOR ethics concerns only"]

**Final Justification:**

The authors provided a thorough response during the rebuttal, and major concerns are addressed.

**Limitations:**

yes

**Paper Formatting Concerns:**

None of any major formatting issues are noticed.

**Quality:**

3

**Strengths And Weaknesses:**

Strengths

- Reformulating RCMDP as a single max‑objective (Eq. 3) eliminates the epigraphic binary search and yields the RNPG update with a principled KL mirror step.
- Theorem 4.1 guarantees an $\epsilon$‑optimal feasible policy in $O(\epsilon^{-2})$ iterations, improving the $O(\epsilon^{-4})$ bound of Kitamura et al. 2024 ([8] in paper).
- Experimental results show 4–8 times faster convergence than EPIRC‑PGS on Garnet and CRS tasks.

Weaknesses
- Assumption 1 requires knowing a constant $\xi > 0$ such that the unknown optimum satisfies all constraints with margin $\xi$. $\xi$ is then used to set other parameters and prove zero‑violation. If $\xi$ is unknown, Theorem 6.1 shows RNPG still converges but only guarantees an $\epsilon$‑level violation and reverts to the same O(\epsilon^{-4}) rate as EPIRC‑PGS. Thus, the headline $O(\epsilon^{-2})$ benefit hinges on oracle information.

- Convergence also needs a “positive‑density” condition on transition kernels, inherited from robust PG theory.
The paper does not quantify how large this density must be, nor test cases where it fails (e.g., sparse transitions), leaving its practical prevalence unclear.

- All experiments use exact tabular robust evaluators; no test shows RNPG with
* sampled value estimates,*
* function approximation,* or
* large state spaces.*
Hence we do not know how the algorithm behaves when the robust oracle is approximate or when KL updates are computed from mini‑batches.

- Every RNPG step still solves $\max_{P\in\mathcal P} V^\pi_P$
for each cost/reward. For KL sets this is closed‑form, but for TV/Wasserstein sets it involves linear programs or iterative solvers. The paper (Alg. 2, appendix) alludes to such a “robust evaluator” but never times it or analyses its complexity; scaling to thousands of states could erase the gains from eliminating the binary search.

Minor comments

- It would be nice to clarify further the meaning of Assumption 2 (p.6) and its practical validity.
- The caption is missing “RNPG” despite the fact that the RNPG curve is plotted.

---

> ### Author Rebuttal · Authors · 2025-07-28
>
> We appreciate your time and thoughtful evaluation of our paper. We recap your comment and present our detailed response as follows. We would also be happy to provide further clarifications if suitable.
>
> >*Assumption 1 requires knowledge of $\xi$..*
>
> * The reviewer is correct. We have also pointed that out in the main text; we will point that out in the Abstract as well. **Also, note that we did not use the feasibility information for our empirical results.**
>
> >*Convergence also needs a “positive‑density” condition on transition kernels..*
>
> * Perhaps, the reviewer is referring to Assumption 2, which states that if the perturbed distribution assigns positive probability to an event, the nominal model should also assign positive probability to that event. *Otherwise, a mismatch in supports could lead to unsampled regions and render finite‑sample bounds intractable*. While this assumption is essential for establishing our theoretical guarantees, we deliberately **did not** enforce it in the empirical study; the algorithm still performed well, suggesting that the practical impact may be less restrictive than the theory implies. Also note that such types of assumptions are common in robust unconstrained PG approaches [A1]. Note that EPIRC-PGS also assumed that the ratio between the state-action occupancy measures on the states covered by all policies and the initial state distribution is bounded.
>
> * Regarding practical prevalence, our experiments include **sparse‑transition** domains such as **RiverSwim**, where many transitions have very small probabilities (Figure 2). In these settings, RNPG **achieves feasibility and outperforms state‑of‑the‑art baselines**, indicating that the method remains effective compared to the state-of-the-art approaches. We will clarify this in the text.
>
> >*All experiments use exact tabular robust evaluators; no test shows RNPG with *sampled value estimates*, *function approximation,* or *large state spaces.*..*
>
> * We now report results in **large/continuous state spaces** using an **actor–critic implementation** with **function approximation** and **sample‑based value estimates**. The robust evaluator is implemented as an **Integral Probability Metric (IPM)‑based** worst‑case critic learned from data (i.e., an approximate robust oracle) inspired by [A1], and RNPG updates are computed using mini‑batches. On benchmarks such as CartPole (and larger state‑space variants), RNPG achieves high returns while satisfying constraints under perturbations, and outperforms other approaches for RCMDP and vanilla CMDP. Further, its wall-clock time is comparable to the non-robust and robust CRPO versions. We will include architectural details, batch sizes, and training protocols in the final version. The results are in the following:
>
> | Algo | Avg. Reward Value | Avg. Cost-value | Wall-Clock Time(s) |
> |---|---|---|---|
> |Vanilla Constrained Actor-Critic with NPG | 323.15  | 350.12 |752.35 |
> |**Robust Constrained Actor-Critic with RNPG (RCAC)** | **298.99** | **199.2**| 770.2 |
> | CRPO | 352.42 | 370.69 |  761.56|
> |Robust CRPO | 281.24 | 179.7 | 701.1|
> |EPIRC-PGS|  114.8 | 53.79 | 1429.14|
>
> For the CartPole environment, we use the perturbation of $\delta=0.1$. The threshold for the constraint is 200. **From the table, it is evident that among all the approaches that are feasible, our approach (RCAC) achieves the best reward.** In particular, it significantly outperforms the EPIRC-PGS both in terms of reward and the wall-clock time.  The main issue with EPIRC-PGS appears to be that the binary search is prone to error, particularly in the function-approximation setup [A2], as the worst-case model might not be exact. We will include the new result in the final version.
>
> * Note that for IPM, there is a closed-form expression of the worst-case value function with the linear representation. We can use that result to achieve the iteration complexity result. However, the complete characterization is left for the future, and we will mention that in the final version.
>
> >*Every RNPG step still solves for each cost/reward. For KL sets this is closed‑form, but for TV/Wasserstein sets it involves linear programs or iterative solvers..*
>
> * For TV/Wasserstein ambiguity sets, computing the worst‑case model/value indeed requires solving a convex program or using an iterative solver. **This requirement, however, is shared by the baselines**—including EPIRC‑PGS and unconstrained robust PG—so the cost of the “robust evaluator” is orthogonal to our contribution. RNPG’s advantage is that it eliminates the outer binary‑search loop, thereby reducing the number of robust‑evaluation calls per update. In other words, when the same evaluator backend is used for all methods, RNPG incurs fewer invocations of that backend and thus reduces end‑to‑end time.
>
> * To make this point concrete, we have added large state‑space experiments where all methods use the same robust‑evaluation module (identical solver and settings). Under this like‑for‑like setup, RNPG achieves better wall‑clock time than EPIRC‑PGS while maintaining feasibility and high reward. We will include full timing tables and implementation details (solver, tolerances, batch sizes) in the revised version.
>
> * For KL ambiguity sets the evaluator admits a closed‑form update. Moreover, for several $L_p$ error‑ball models, the worst‑case model/value also has closed‑form expressions (see [A3]), making RNPG directly applicable with low evaluator overhead.
>
> >*It might have been interesting to also see a baseline of a naive Lagrange method or something like CRPO applied with worst-case values...*
>
> * Thank you for the suggestion. We have added a **robust CRPO** baseline that uses the same worst-case evaluator as RNPG (i.e., robust values under the chosen ambiguity set) for the constrained Garnet environment (Section 5, Figure 1). The results are in the followings:
>
> | Algo | Avg. Reward Value | Avg. utility-value | Wall-Clock Time (s) |
> |---|---|---|---|
> |EPIRC-PGS | 108.25  | 107.33 | 390.21 |
> |**RNPG** | **170.142** | **116.55**| **99.21**|
> | RPPG | 118.54 | 107.2 | 103.21 |
> | Robust- CRPO | 142.89 | 101.05 | 101.51 |
>
> Here, the **utility** needs to be above $100$ for feasibility. Again, RNPG outperforms EPIRC-PGS and Robust-CRPO in terms of reward while achieving feasibility. Further, its wall-clock time is $4\times$ smaller than the EPIRC-PGS. We will add the details in the final version.
>
> [A1]. Zhou, Ruida, et al. "Natural actor-critic for robust reinforcement learning with function approximation." Advances in neural information processing systems 36 (2023): 97-133.
>
> [A2]. Waeber, Rolf, Peter I. Frazier, and Shane G. Henderson. "Bisection search with noisy responses." SIAM Journal on Control and Optimization 51.3 (2013): 2261-2279.
>
> [A3]. Kumar, Navdeep, et al. "Policy gradient for rectangular robust markov decision processes." Advances in Neural Information Processing Systems 36 (2023): 59477-59501

---

> > ### Author Response · Authors · 2025-08-06
> >
> > Dear Reviewer GbJz,
> >
> > We would like to thank you for your time and thoughtful evaluation of our paper. We are glad that you positively evaluate our contributions. Since the author-reviewer discussion period is ending soon, and we have not heard from you, we just wanted to check in and ask if the rebuttal clarified and answered the questions raised in your review. We would be very happy to engage further if there are additional questions!

---

> > > ### Comment · Reviewer_GbJz · 2025-08-07
> > >
> > > Thank you. I do not have more questions.

---

> ### Author Response · Authors · 2025-08-08
>
> We are glad that our responses have resolved your questions, and we are thankful for your positive evaluation.

---

### Decision · Program_Chairs · 2025-09-17

**Decision:**

Accept (poster)

**Comment:**

This paper presents an algorithm, Robust Natural Policy Gradient (RNPG), for solving robust constrained Markov decision processes (RCMDPs). The key innovation is a reformulation of the RCMDP objective that circumvents the need for the computationally expensive binary search, which is required by prior state-of-the-art methods. The authors provide strong theoretical guarantees. The reviewers unanimously acknowledged the significance of this contribution, highlighting the principled approach of eliminating the binary search and the improved theoretical bounds. The author's rebuttal successfully address the reviewers' initial concerns.